SciPost Physics

Submission

# Anisotropy-mediated reentrant localization

X. Deng[1*], A. L. Burin[2], and I. M. Khaymovich[3,4],

**1** Institut für Theoretische Physik, Leibniz Universität Hannover, Appelstr. 2, 30167 Hannover, Germany
**2** Department of Chemistry, Tulane University, New Orleans, Louisiana 70118, USA
**3** Max-Planck-Institut für Physik komplexer Systeme, Nöthnitzer Straße 38, 01187-Dresden, Germany
**4** Institute for Physics of Microstructures, Russian Academy of Sciences, 603950 Nizhny Novgorod, GSP-105, Russia
* Xiaolong.Deng@itp.uni-hannover.de

July 21, 2021

## Abstract

We consider a 2d dipolar system, $d = 2$, with the generalized dipole-dipole interaction $\sim r^{-a}$, with the power $a$ controlled experimentally in trapped-ion or Rydberg-atom systems via their interaction with cavity modes. We focus on the dilute dipolar excitation case when the problem can be effectively considered as single-particle with the interaction providing long-range dipolar-like hopping. We show that the spatially homogeneous tilt $\beta$ of the dipoles giving rise to the anisotropic dipole exchange leads to the non-trivial reentrant localization beyond the locator expansion, $a < d$, unlike the models with random dipole orientation. The Anderson transitions are found to occur at the finite values of the tilt parameter $\beta = a$, $0 < a < d$, and $\beta = a/(a - d/2)$, $d/2 < a < d$, showing the robustness of the localization at small and large anisotropy values. Both extensive numerical calculations and analytical methods show power-law localized eigenstates in the bulk of the spectrum, obeying recently discovered duality $a \leftrightarrow 2d - a$ of their spatial decay rate, on the localized side of the transition, $a > a_{AT}$. This localization emerges due to the presence of the ergodic extended states at either spectral edge, which constitute a zero fraction of states in the thermodynamic limit, decaying though extremely slowly with the system size.

# 1  Introduction

With the realization of Anderson localization [1] of matter waves in optical lattice and of light [2], many extensions of disordered quantum systems are proposed [3] and implemented with and without interactions. A few of notable examples are vibrational modes of polar molecules [4], Rydberg atoms [5,6], nitrogen vacancy centers in diamond [7], magnetic atoms [8, 9], photonic crystals [10], nuclear spins [11], trapped ions [12,13] and Frenkel excitations [14].

In all these systems power-law interactions of dipole-dipole kind are ubiquitous [3]. In addition, in the experiments of atomic systems the exponent $a$ of this power-law decay can be precisely controlled in a wide range, $0 < a < 2$ [12, 13] and for $a = 3$ or $a = 6$ [5,6]. If the excitations in such systems are dilute, the dipole-dipole interaction induces dipole flips of far-away dipoles. Thus, this problem has an effective single-particle description of (nearly) non-interacting dipole excitations, where the above interacting term works as the power-law decaying off-diagonal dipole-flip hopping. In $2d$ setting, for homogeneous orientation of all dipoles, perpendicular to their plane, the corresponding disordered model has *deterministic isotropic* long-range hopping. Recent studies show that such models in the dimensionality $d$ with fully-correlated hopping terms are localized even beyond the locator expansion convergence [15–18]. In particular, isotropic power-law hopping models $1/r^a$ show the power-law localization with the duality between the perturbative regime, $a < d$, and beyond it, $a > d$ [16,17]. For all $d \le 2$ only the measure zero of the states located at one the spectral

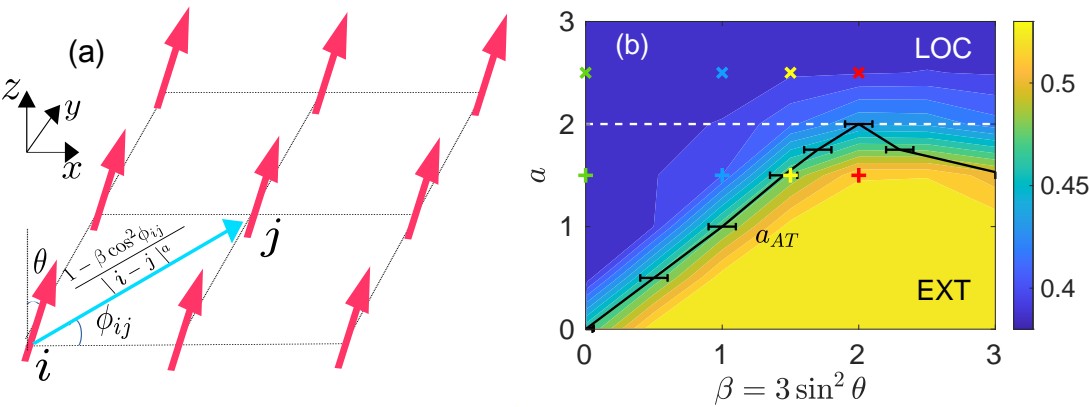

Figure 1: **Model and phase diagram.** (a) Two-dimensional (2d) lattice of quantum dipoles with dipole-dipole anisotropic interaction $(1 - \beta \cos^2 \phi_{ij})/|i - j|^a$, and the generalized power-law decay exponent $a > 0$. $i = (i_x, i_y)$ is the coordinate vector of $i$th dipole. The anisotropy parameter $\beta = 3 \sin^2 \theta$ is governed by the homogeneous tilt angle $\theta$ of all dipoles from the normal $z$-axis to the lattice plane. $\phi_{ij}$ is the angle between the spatial 2d vector $i - j$ and the $x$-axis-aligned electric field. (b) The phase diagram of the anisotropic 2d dipole model with dilute excitations and the on-site disorder. The color plot shows the $r$-statistics at $L = 200$ (see main text), the black solid line $a = a_{AT} = \min [\beta, \beta/(\beta - 1)]$ separates the localized ("LOC") phase, $a > a_{AT}$, from the extended phase, $a < a_{AT}$, while the black error bars correspond to these phase transitions, extracted from its finite-size scaling, Fig. 3. According to that analysis the transition occurs at $r \simeq 0.47$. The selected points with symbols "+" and "×" of the same colors (used in further figures) indicate the duality of power-law localization of wave functions for $a < d = 2$ and $a > d$.

edges might be delocalized in such models.

However, experimentally feasible dipolar systems are also characterized by common anisotropy which may have drastically different physics from the isotropic case. Usually the anisotropic terms are considered as quasi-disorder [19–22] and in the case of random and heterogeneous dipole orientations they lead to the localization-delocalization transition at $a = d$. In this paper we show that the situation is more subtle in the case of homogeneous dipole anisotropy, relevant for the experiments in the electric field, Fig. 1(a). This anisotropy gives rise to the *reentrant* localization phase diagram, Fig 1 (b) beyond the locator expansion, $a < d$.

In order to combine measure zero of delocalized states for the isotropic case with the possibility of anisotropy, we focus on a two-dimensional, $d = 2$, quantum dipolar system with the on-site disordered chemical potential and add a spatially homogeneous angular anisotropy, Fig. 1(a). We show that the Anderson localization beyond the locator expansion is robust to the homogeneous tilt, $\beta = 3 \sin^2 \theta$, of all dipoles up to a finite critical tilt value, Fig. 1(b), unlike the models with uncorrelated random off-diagonal hopping (see, e.g., [23]). Moreover, we demonstrate that the anisotropy leads to the reentrant character of localization showing localized eigenstates both at small (nearly isotropic) and large (strongly anisotropic) tilt. Such systems bridge the gap between models with deterministic and random interactions and bring new dimensions of anisotropy-mediated localization to the field of long-range systems. The extensive numerical simulations showing consistent behavior of level statistics and spatial wave-function properties are analytically supported by the renormalization group analysis (similar

to [15, 18]) and the newly developed matrix inversion trick [17].

## 2   Models and methods

We consider the model describing dilute polar excitations propagating via dipole-flips (induced by their dipole-dipole interaction) on a square lattice of sites $\{i = (i_x, i_y)\}$ of size $L$, $i_x, i_y = 0, 1, \ldots, L - 1$, Fig. 1(a), with the Hamiltonian

$$H = -\sum_{i,j} \frac{1 - \beta \cos^2 \phi_{ij}}{r_{ij}^a} |i\rangle\langle j| + \sum_i \mu_i |i\rangle\langle i|, \tag{1}$$

where $\{|i\rangle\}$ are site basis states, $\mu_i \in [-\frac{W}{2}, \frac{W}{2}]$ is on-site disorder uniformly distributed over the above interval, the hopping term depends on the distance $r_{ij} = \sqrt{(i_x - j_x)^2 + (i_y - j_y)^2}$ between two lattice sites and its angle $\phi_{ij}$ with respect to electric field $x$-axis. The effective single-particle hopping model (1) is obtained from the model of dipoles with dipole-dipole interactions as these interactions induce effective anisotropic transfer of excitations between sites via dipole-flips (see, e.g., [19, 24]). The anisotropy parameter $\beta = 3 \sin^2 \theta$ is introduced by analogy to the experimental setup of dipolar molecules, Fig. 1(a), and is related to the homogeneous tilt angle $\theta$ of dipoles w.r.t. the $z$-axis. In this work we restrict our consideration to the physical values of $0 \le \beta \le 3$. The isotropic limit, $\beta = 0$, considered in a pioneering paper by Burin and Maksymov [15] for $a = d = 3$ and investigated in details for $d = 1$ in [16, 17] represents a newly discovered universality class of long-range models with fully-correlated hopping. It is these complete correlations that allow destructive interference of long-range hops, similarly to the standard weak and Anderson localization case, and localize the bulk of the system for all values of $a$ at $d \le 2$.

In the opposite limit of a long-range model with *fully uncorrelated random-sign* hopping $h_{ij}/r_{ij}^a$ [19–21, 25, 26] it is well-known that the localization occurs only for $a > d$, while the ergodic delocalization spans over the entire range $a < d$. The pure $d$-dimensional dipolar case of our model, $\beta = d$, (initially considered in [19–21, 27] for different $d$) leads to the same result, see Fig. 1(b).

One may naively expect that the intermediate case of $\beta \ne 0, d$ is similar to the perturbation of the fully-correlated model ($\beta = 0$) by a fraction $\epsilon \sim \beta/d$ of random-sign hopping $(1 + \epsilon h_{ij})/r_{ij}^a$ (at least for $0 < \beta < d$) as finite $\beta$ works as a kind of quasi-disorder. However, in the latter model any $\epsilon > 0$ immediately delocalizes all the spectral states at $a < d$ as shown in [16, 23], which is not consistent with the phase diagram, Fig. 1(b).

Instead, in the anisotropic model (1) there is a *finite* tilt $\beta_{AT}(a)$ until which the Anderson transition survives

$$\beta_{AT}(a) = a \quad \Leftrightarrow \quad a_{AT}(\beta) = \beta \,, \quad 0 \le \beta_{AT}, a \le 2 \,. \tag{2}$$

This is the main result of the paper summarized in the phase diagram, Fig. 1(b). The Hamiltonian (1) obeys the $\pi/2$-rotational symmetry of a square lattice, $\phi_{ij} \leftrightarrow \phi_{ij} + \pi/2$, combined with the disorder strength and the tilt rescaling

$$W \leftrightarrow \frac{W}{1 - \beta} \,, \quad \beta \leftrightarrow \frac{\beta}{\beta - 1} \,. \tag{3}$$

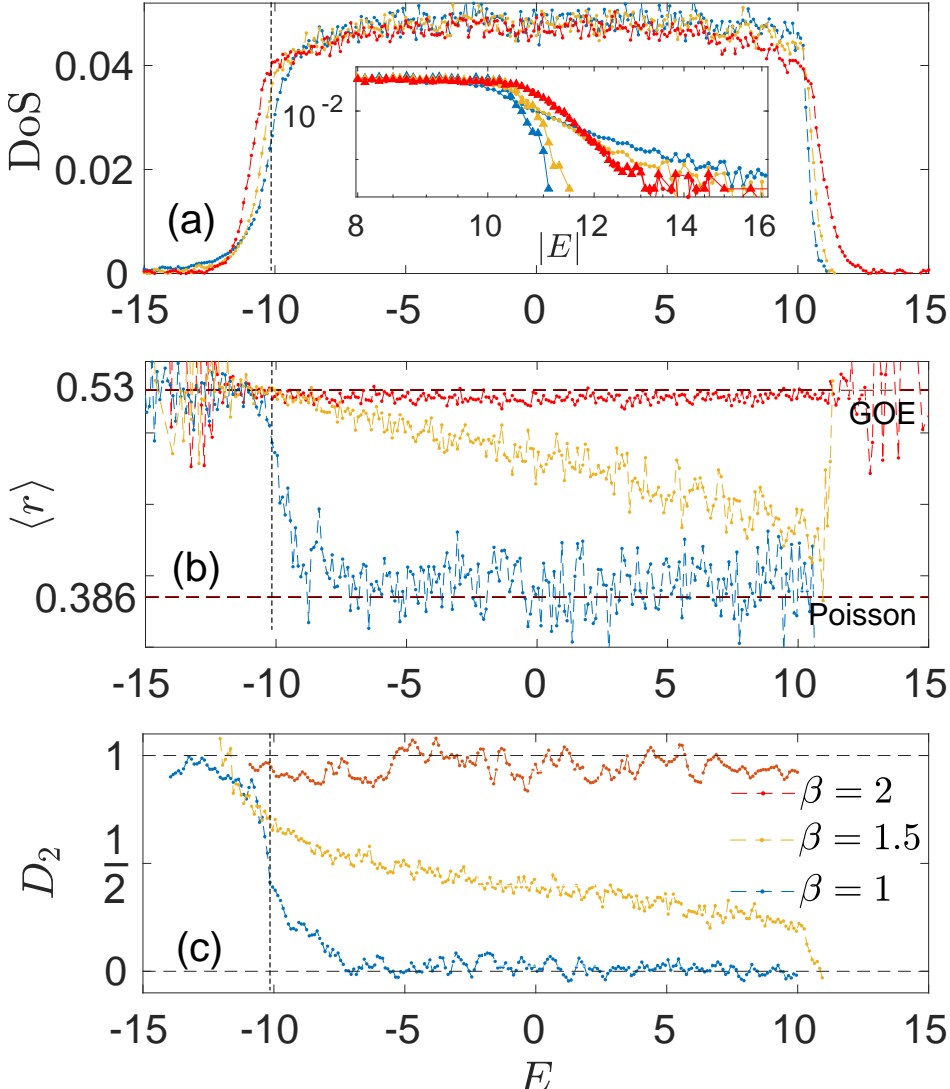

Figure 2: **Emergence of the finite-size mobility edge across the Anderson transition.** (a) global density of states (DOS), (b) level-spacing ratio $r$-statistics, and (c) fractal dimensions $D_2$ for each eigenstate versus energy $E$ in the localized ($a = 1.5$, $\beta = 1$, blue), critical ($a = \beta = 1.5$, yellow), and delocalized ($a = 1.5$, $\beta = 2$, red) phases. Both panels (b) and (c) show localized, multifractal and ergodic eigenstate properties in the spectral bulk. The bulk states are within the range $[-W/2, W/2]$, where we take $W = 20$. The inset to panel (a) shows power-law tails of DOS at either ($a > a_{AT}$) or both ($a < a_{AT}$) spectral edges. The data for $D_2$ are extrapolated from $L = 100$, 150, 200, and 250 with the corresponding number of disorder realizations 1000, 500, 100, and 50, respectively, see Appendix A.2 for details. For the rest of the data $L = 250$. The vertical dashed lines provide the position of the FSME extracted from the finite-size data for $r$-statistics.

This symmetry relates the interval $0 < \beta < 2$ to the ones $\beta < 0$ and $\beta > 2$ and causes the reentrant character of the above phase diagram. Further without loss of generality we restrict ourselves to $0 < \beta < 2$.

The phase diagram, Fig. 1(b), showing the localization properties of the bulk states, is obtained from extensive numerical simulations.

## 3 Methods

The eigenfunctions $\psi_n(i)$ and eigenenergies $E_n$ of the Hamiltonian, Eq. (1), are numerically calculated by exact diagonalization for 2d square samples of the linear size $L$ from 75 to 280 and for $10^2 - 10^3$ random realizations of the diagonal disorder. The ratio level statistics, Figs. 1(b) and 2(b),

$$r = \left\langle \min\left(r_{n,1}, \frac{1}{r_{n,1}}\right)\right\rangle \ , \quad r_{n,1} = \frac{E_n - E_{n-1}}{E_{n+1} - E_n} \tag{4}$$

is calculated across the entire spectrum. It shows the Poisson value $r = 2\ln 2 - 1 \simeq 0.3863$ for all spectral bulk states in the localized phase, and $r \approx 0.5307$ of Gaussian orthogonal ensemble (GOE) [28, 29] at the spectral edge and for all eigenstates in the extended phase. The more detailed analysis of the finite-size scaling (FSS) of $r$-statistics determines the transition line $\beta = \beta_{AT}(a)$, Eq. (2), via the change of finite-size flow of $r(a, \beta, L)$ versus $L$, Fig. 3(a) and the black error bars in Fig. 1(b), see Appendix A.1 for the calculation details. The standard FSS collapse $r = R\left[(\beta - \beta_{AT})L^{1/\nu}\right]$ gives $\beta_{AT} = a \pm 0.1$ for $0 < \beta < 2$ and $\nu = 1.0 \pm 0.2$ for all considered $a$, Fig. 3(b). At the transition line the $r$-statistics takes the universal value $\langle r \rangle \approx 0.47$ independent of $a$. The fractal dimension $D_2$ extracted from the inverse participation ratio $I_2 = \sum_i |\psi_n(i)|^4 \propto L^{-dD_2}$ shows consistent behavior in the localized ($D_2 \to 0$), critical ($0 < D_2 < 1$), and extended phases ($D_2 \to 1$), Fig. 2(c). However, in the latter case the finite-size data converge to 1 very slowly and the higher-order extrapolation in $1/\ln L$ or the linear one with irrelevant exponent [30] shows significant fluctuations, see [31] and Appendix C for details.

From the above mentioned measures one can extract the position of the finite-size mobility edge (FSME) $E^*$ below which all the states are ergodic, while being power-law localized above it for $a > a_{AT}$ and extended with smaller extrapolated $D_2$ for $a < a_{AT}$. The FSS, the inset to Fig. 3(b), shows that the corresponding fraction of ergodic states $f_{\text{erg}} = \int_{E < E^*} \rho(E)dE/L^d$ in the localized phase decays with the system size $L$, but does it logarithmically slowly, see Appendix A.3. This leads us to the conclusion that in the localized phase, $a > a_{AT}$, there is measure zero of the delocalized edge states which can be neglected in the thermodynamic limit.

The non-trivial phase diagram for the bulk spectrum, Fig. 1(b), and anisotropy-mediated reentrant localization can be understood from the atypical extended nature of high-energy states in both isotropic, $\beta = 0$, and anisotropic, $\beta > 0$ cases of a dipole system. Indeed, the hopping term $\sum_{\mathbf{q}} V_{\mathbf{q}}|\mathbf{q}\rangle\langle\mathbf{q}|$ of (1), diagonal in the momentum basis $|\mathbf{q}\rangle = \sum_n e^{i\mathbf{q}\mathbf{n}}|\mathbf{n}\rangle/L^{d/2}$ due to its translation invariance, diverges at small $|\mathbf{q}| < q_* \ll 1$ and $a < d = 2$

$$V_{\mathbf{q}} = -\int_0^\infty r dr \int_0^{2\pi} d\phi e^{iqr\cos(\phi-\phi_{\mathbf{q}})} \frac{1 - \beta\cos^2\phi}{r^a} = c_a q^{a-2}\left[\beta - a - (2-a)\beta\cos^2\phi_{\mathbf{q}}\right] \ . \tag{5}$$

Here $c_a = -\frac{\pi\Gamma(-a/2)}{2^a\Gamma(a/2)} > 0$, $\Gamma(a)$ is a Gamma-function, and the momentum $\mathbf{q} = \frac{\pi}{L}(m_x, m_y) = q(\cos\phi_{\mathbf{q}}, \sin\phi_{\mathbf{q}})$ is written in polar coordinates $q$, $\phi_{\mathbf{q}}$, with $m_x, m_y = 0, 1, \ldots, L - 1$, see Appendix D for the calculation details.

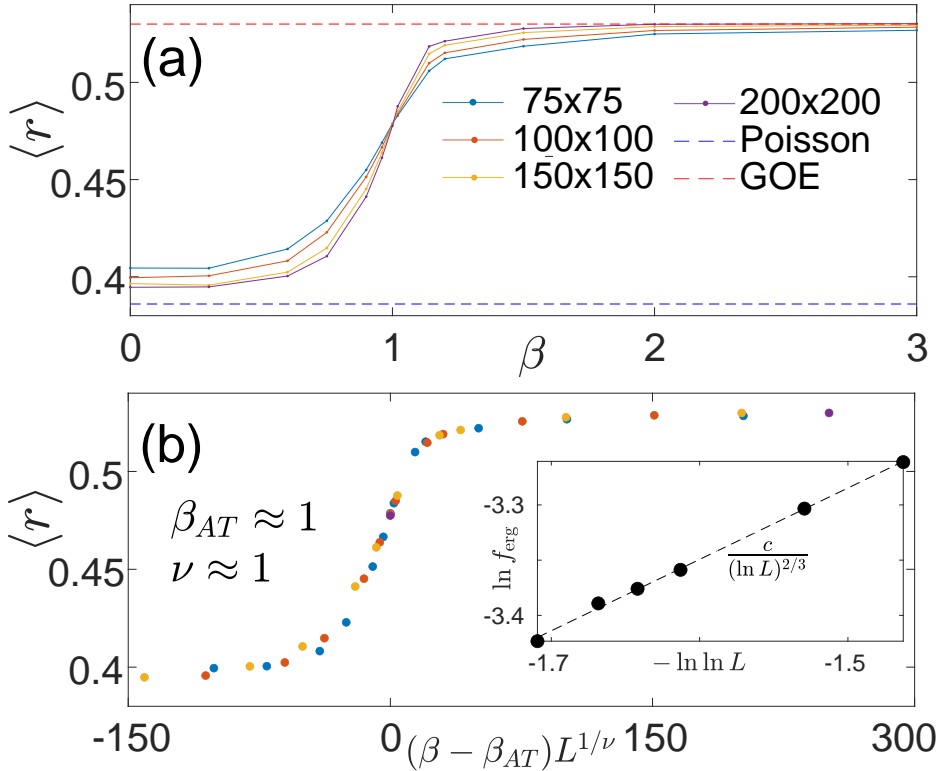

Figure 3: **Anisotropy-driven Anderson transition and $r$-statistics collapse.** (a) Level statistics versus anisotropy $\beta$ at $a = 1$ at system sizes shown in legend; (b) Scaling collapse of $\langle r \rangle$ using $\langle r \rangle = R\left[(\beta - \beta_{AT})L^{1/\nu}\right]$, giving $\beta_{AT} = 1.0 \pm 0.1$ and $\nu = 1.0 \pm 0.2$ for $a = 1$. The calculations of $\langle r \rangle$ are done on eigenstates from the interval $E \in [-W/2, W/2]$, $W = 20$, which constitute 95% of all the states. (inset) Fraction $f_{\text{erg}}$ of ergodic extended states below the finite-size mobility edge, $E < E^*$, in the localized phase, $a = 1.5$, $\beta = 0.3$, versus the system size $L$. The number of disorder realizations is from 2000 for $L \leq 100$ to 150 for $L = 250$. Numerical $f_{\text{erg}}$ (symbols) is consistent with analytical predictions (dashed line) in (8).

Thus, there are large-negative-energy eigenstates, $E_{\mathbf{q}} \simeq V_q < E^* < 0$, with $|E_{\mathbf{q}}| \gg W$, which are barely affected by the on-site disorder and, thus, are diffusive or ballistic states represented by superpositions of plane waves only with small momenta $q < q_*$. Although the above *exact* eigenstates $E_q \simeq V_q < E^* < 0$ constitute a *zero* fraction of all states, they give the dominant contribution to the hopping term

$$\sum_{\mathbf{q}} V_{\mathbf{q}} |\mathbf{q}\rangle \langle \mathbf{q}| = \sum_{E_{\mathbf{q}} < E^*} E_{\mathbf{q}} |E_{\mathbf{q}}\rangle \langle E_{\mathbf{q}}| + J_{\text{res}} \ . \tag{6}$$

The action of this term on the bulk eigenstates $E_n > E^*$, being orthogonal to the above ones, $\langle E_{\mathbf{q}} | E_n \rangle = 0$, is equivalent to the residual hopping term $J_{\text{res}}$ with substantially suppressed spatial structure. This short-range effective hopping leads to the localization of the entire spectral bulk providing a new localization mechanism due to the presence of measure zero of delocalized high-energy states orthogonal to them [1]. These simple arguments work provided

---

[1]Similar effects have been recently observed in non-integrable many body systems where the special spectral-edge states lead to the departure from the eigenstate thermalization hypothesis in the spectral bulk [32].

the extended high-energy states appear on the *only* spectral edge and, thus, their contribution to (6) is compensated by the states from the opposite one. The effect of extended spectral edge states has been partially understood for the case of the only such state in terms of cooperative shielding in [33] and explained in details for the general case $a \geq 0$, $d = 1$ by the matrix inversion trick in [17, 23] and by the renormalization group in [18].

In our model (1), if $V_{\mathbf{q}}$ do not change the sign with $\phi_{\mathbf{q}}$ in order to have high-energy states on the *only* spectral edge,

$$V_{\mathbf{q}}(\phi_{\mathbf{q}})/V_{\mathbf{q}}(0) > 0 \quad \Leftrightarrow \quad a|\beta - 2| > |a - 2|\beta , \tag{7}$$

it immediately provides the phase boundary of the localization $\beta < \beta_{AT}(a)$, Eq. (2), valid for all $a$ and $\beta$. The power-law growth of the spectral-edge energies $E_{\mathbf{q}} \simeq V_{\mathbf{q}} \sim q^{a-2}$ with decreasing momentum $q$ is explicitly represented by the power-law decaying tail of DOS on either (both) spectral edge(s) in the localized (extended) phase, see the inset to Fig. 2(a).

The FSME $E^* \simeq V_{\mathbf{q}_*}$ found numerically can be determined by Ioffe-Regel criterion, see Appendix F for details. Indeed, a state is localized as soon as its localization length $\ell_{loc}$ is smaller than the system dimension $L$. In 2d systems the localization length is exponentially growing with the mean-free path $\ell_{loc} \sim e^{ck_F \ell_{mfp}}$, with $c \sim O(1)$. Fixing the Fermi momentum at $k_F = q$ one calculates the mean-free path $\ell_{mfp}(q) \simeq v_q \tau_q$ via the group velocity $v_q = dV_q/dq \sim q^{a-d-1}$ at the momentum $q$ and the level broadening determined by the Fermi Golden rule $\gamma_q = \tau_q^{-1} \sim W^2 \rho(E_{\mathbf{q}}) \sim W^2 q^{2d-a}$ for the plane wave scattering on impurities $\mu_i \simeq W$. This gives the fraction $f_{\mathrm{erg}}$ of ergodic extended states below the FSME, $\ell_{loc} \sim L$,

$$f_{\mathrm{erg}} = \pi q_*^2 \sim \left[ W^2 \ln L \right]^{-1/(3-a)} , \tag{8}$$

which decays only as a power of the logarithm of the system size, see inset to Fig. 3(b).

Focusing now on the properties of the bulk spectral states we investigate them in terms of the multifractal analysis and spatial decay in more details. Indeed, the multifractal spectrum (MFS) $f(\alpha)$, characterizing the multifractality of the states, is defined by the probability distribution $\mathcal{P}(\ln |\psi_n(i)|^2) \sim L^{d(f(\alpha)-1)}$ of the logarithm of the wave-function intensity $\alpha = -\ln |\psi_n(i)|^2 / \ln L^d$ [26] and can be extracted directly from the histogram over $\alpha$ [34–36]. For the non-ergodic extended states in most cases the MFS obeys a so-called Mirlin-Fyodorov symmetry $f(1 - \delta\alpha) = f(1 + \delta\alpha) - \delta\alpha$ [26]. The ergodic extended state corresponds to a $\delta$-function at $\alpha = 1$, while the localized state has $f(0) = 0$ and a certain (usually linear) form of $f(\alpha > 0) = k\alpha$, with $k = 0$ for exponential and $k > 0$ for power-law localization. In the model (1) the multifractal spectrum of the bulk spectral states, Fig. 4(a), shows power-law localized ($\beta = 1$), multifractal ($\beta = 1.5$), and ergodic ($\beta = 2$) behavior in the localized phase, at the transition, and in the extended phase, respectively.

The spatial decay $\langle \ln |\psi_n(i)|^2 \rangle$ of the typical wave-function intensity $|\psi_n(i)|^2$ with the distance $r = |i - i_0|$ from its maximum $i = i_0$ suggested as the localization measure in [16] and used in [17, 18, 23] shows the same duality of the power-law decay rate

$$|\psi_n(i)| \sim r^{-a} \text{ for } a > d \tag{9a}$$

$$|\psi_n(i)| \sim r^{a-2d} \text{ for } a < d \tag{9b}$$

as in [16–18] in the whole range of anisotropy parameter $\beta$ in the localized phase (blue lines) in Fig. 4(b) and Appendix B. In the extended phase, eigenstates develop the dual power-law decay at distances, small compared to the sample size, but become ergodic at larger distances,

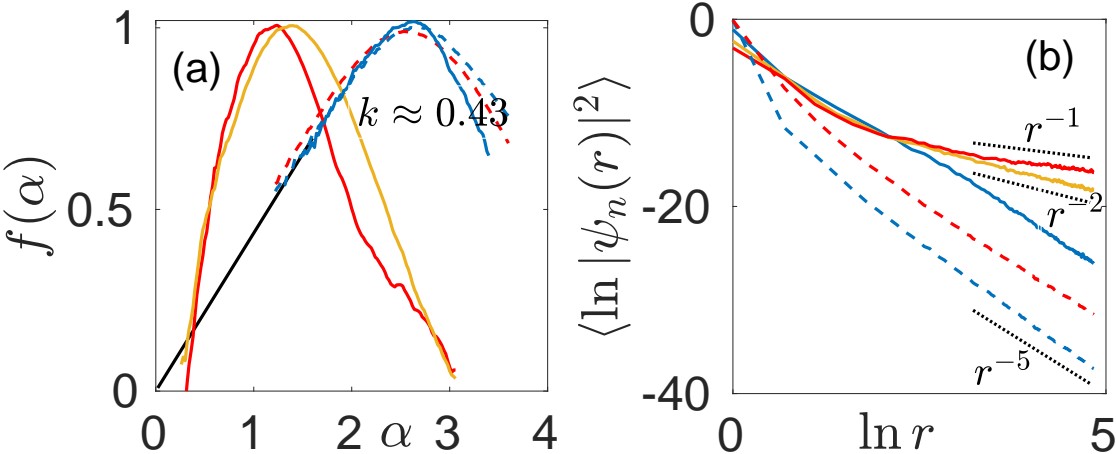

Figure 4: **Spatial properties in the spectral bulk.** (a) spectrum of fractal dimensions $f(\alpha)$ and (b) power-law spatial decay of eigenstates in the bulk of the spectrum for $a = 1.5$ (solid), 2.5 (dashed), and $\beta = 1$ (blue), 1.5 (yellow), 2 (red). The panel (b) confirms the duality of power-law spatial decay rate $\gamma(a) \approx \gamma(2d - a)$ [16–18] in the localized phase ($\beta < a < d$ or $a > d$), also supported by the slope $k < 0.5$ of $f(\alpha)$ in panel (a). $f(\alpha)$ is extrapolated from $L = 75, 100, 125, 150, 200, 225$ and $250$ with the corresponding number of disorder realizations from 2000 for $L \leq 125$ to 300 for $L \geq 225$, see Appendix A.2 for details. and with the disorder amplitude $W = 20$. For the spatial decay $L = 250$ and $W = 20$ for $a = 1.5$, for $a = 2.5$ we choose bigger $W = 200$ in order to make the power-law tail dominant on moderate sizes.

see Appendix C for details. Thus, we have to conclude ergodicity in this phase with strong finite-size effects.

At the self-dual line $a = d = 2$ of (9) the wave-function behavior is consistent with the critical localization, Fig. 5, $f(\alpha) \simeq k\alpha$, with $k = 1/2$ corresponding to the localized eigenstate and the spatial decay [18]

$$|\psi_n(i)| \sim r^{-d} (\ln r)^{-2} \ . \tag{10}$$

The pure 2d dipole point $a = \beta = 2$ considered in [27] and revisited in [31] is exempted here as it shows the transition from ergodicity to localization over the disorder amplitude.

Both (9) and (10) can be understood in terms of the renormalization group (RG) analysis similar to the one in [15, 18].

## 4   Renormalization group analysis

The main assumption of this RG written in the limit of large disorder strength $W \gg 1$ is that at each step over the hopping radius $R$, taken into account at this step, the localization length $\ell^{(R)}$ of an eigenstate $|\psi_n^R\rangle = \sum_i \psi_n^R(i) |i\rangle$ around its maximum $i = n$ is small compared to $R \gg \ell^{(R)}$. This allows to take into account only resonant site pairs and approximate the renormalized hopping potential as follows

$$\sum_{i,j} \frac{1 - \beta \cos^2 \phi_{ij}}{r_{ij}^a} |i\rangle\langle j| \simeq \sum_{n,m} l_m l_n \frac{1 - \beta \cos^2 \phi_{mn}}{r_{mn}^a} |\psi_m^R\rangle\langle\psi_n^R| \ , \tag{11}$$

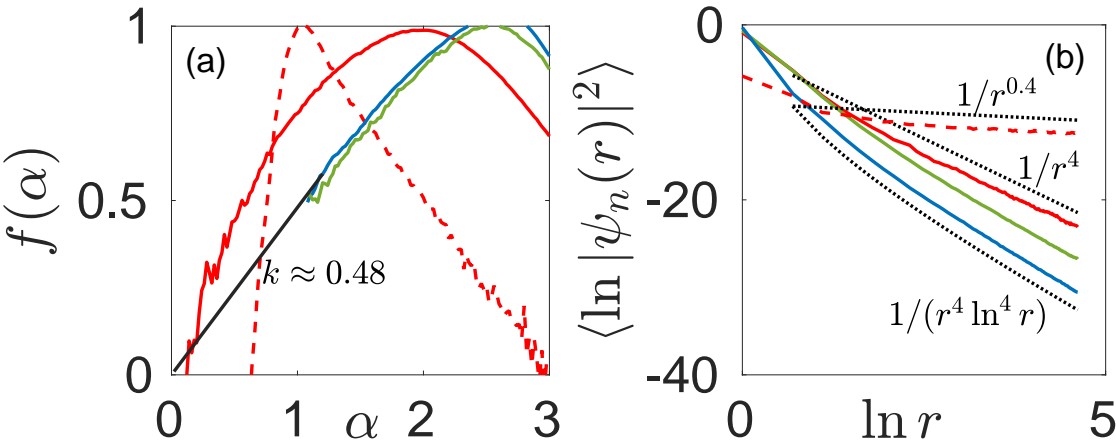

Figure 5: **Spatial properties at the dual line** $a = d = 2$. (a) spectrum of fractal dimensions $f(\alpha)$ and (b) power-law spatial decay of eigenstates in the bulk of the spectrum at the self-dual line $a = 2$ of (9) for $\beta = 1$ (blue), 2 (red), 3 (green). The linear behavior of $f(\alpha)$ with the slope close to $k = 0.5$ supports the critical localization for $\beta \neq 2$. The exceptional point $a = \beta = 2$ shows the transition from ergodicity ($W = 4$, dashed) to localization ($W = 40$, solid) over the disorder amplitude. The disorder strength for $\beta = 1, 3$ is $W = 40$. $f(\alpha)$ is extrapolated from $L = 75$, 100, 125, 150, 200, 225 and 250 with the corresponding number of disorder realizations from 2000 for $L \leq 125$ to 100 for $L = 250$, see Appendix A.2 for details. For the spatial decay $L = 200$.

with $l_n = \sum_i \psi_n^R(i)$. Here we neglect the difference between bare hopping terms at the distance $r_{ij}$ and $r_{mn}$ due to smallness of $|r_{ij} - r_{mn}| < r_{in} + r_{jm} < 2\ell^{(R)} \ll R$. Please see Appendix E for more details.

The upper estimate of the renormalization prefactor $l_m l_n$ at a certain energy $E_n, E_m \simeq E$ can be written as

$$\langle l^2 \rangle_E = \frac{\langle \sum_n l_n^2 \delta(E - E_n) \rangle}{\rho(E)} \simeq \frac{\sum_{|\mathbf{m}-\mathbf{n}|<R} \langle \operatorname{Im} G_{\mathbf{m}-\mathbf{n}} \rangle}{\pi \rho(E)} \simeq \frac{\operatorname{Im} \bar{G}_{|q| \simeq 1/R}(E)}{\pi \rho(E)}, \qquad (12)$$

via DOS and the Green function $\bar{G}_{\mathbf{q}}(E)$ averaged of the on-site disorder. The latter reads as $\bar{G}_{\mathbf{q}}(E) = [E - V_{\mathbf{q}} - \Sigma]^{-1}$, with the self-energy given by a simplest coherent potential approximation $\Sigma = -\frac{W^2}{12} \bar{G}_0(E)$, consistent with the Fermi Golden rule result $\operatorname{Im} \Sigma = -\gamma_{\mathbf{q}}$. In the spectral bulk $E \sim W$, DOS is determined by the disorder $\rho(E) \sim 1/W$ and thus

$$\langle l^2 \rangle_E = \frac{W}{2\pi^2} \operatorname{Im} \int_0^{2\pi} \frac{d\phi_{\mathbf{q}}}{E - V_{\mathbf{q}} - \Sigma} = \frac{W}{2\pi^2} \int_0^{2\pi} \frac{\gamma_{\mathbf{q}} d\phi_{\mathbf{q}}}{(E - V_{\mathbf{q}} - \operatorname{Re} \Sigma)^2 + \gamma_{\mathbf{q}}^2} . \qquad (13)$$

At $a < d$ the integrand denominator dominated by the hopping spectrum, $V_{\mathbf{q}}$ has infrared divergence, so the angle averaging depends on whether $V_{\mathbf{q}}$ versus $\phi_{\mathbf{q}}$ changes the sign or not for $q \simeq 1/R \ll 1$.

For sign-definite $V_{\mathbf{q}}$, Eq. (7), the integral is given mostly by $\langle l^2 \rangle_E \sim W\gamma_{\mathbf{q}}/V_{\mathbf{q} \simeq 1/R}^2 \sim W^2 R^{2(a-d)}$ and leads to (9b). This result can be equivalently obtained from the matrix-inversion trick [17]. More rigorous calculations done at $a = d = 2$ [18] give logarithmic corrections leading to (10). In the opposite case of $a < a_{AT}$, $V_{\mathbf{q}}$ changes sign w.r.t. $\phi_{\mathbf{q}}$ and

simple calculations give $\langle l^2 \rangle_E \sim W/V_{q \simeq 1/R} \sim R^{a-d}$ resulting in $|\psi_E(i)|^2 \sim r^{-d}$. This critical behavior, formally equivalent to the critical case of $a = d$ for the random-sign hopping term $h_{ij}/r_{ij}^a$, hints that the delocalized phase at $a < a_{AT}$ is nonergodic. However, more rigorous calculations of transport based on Kubo formula [31] give logarithmic corrections leading to ergodic behavior. This analysis puts the ground to the simple localization-delocalization arguments given between Eqs. (6) and (7) about the presence of high-energy states on either or both spectral edges.

# 5 Conclusion and Outlook

To sum up, we explicitly show both numerically and analytically the phenomenon of the anisotropy-mediated reentrant Anderson localization transition relevant for generic 2d quantum dipole models. The transition is demonstrated to occur at a finite anisotropy tilt angle of dipoles depending on the exponent $a$ of the generalized dipole-dipole interaction controlling excitation hopping. Moreover, close to the pure 2d dipole-dipole interaction $1 < a \leq 2$ the phase diagram has a reentrant nature showing the localization both at large and small tilts.

The further research should take into account the robust localized nature of eigenstates in dilute dipolar systems with respect to the ones with randomized interaction strength. This difference between the models with deterministic and random interactions plays an important role also in dense systems where the many-body localization has different properties for such systems (see, e.g. [37–49]). It might be interesting to understand whether there is a many-body localization transition driven by the anisotropy of long-range couplings in such systems.

## Acknowledgements

We thank V. E. Kravtsov for illuminating discussions. We also thank G. V. Shlyapnikov and Luis Santos for previous collaboration on related topics.

**Funding information** This research was supported by the DFG projects SA 1031/11, SFB 1227 DQ-mat (X. D.), by Russian Science Foundation, Grant No. 21-12-00409 (I. M. K.), by Carrol Lavin Bernick Foundation Research Grant (2020-2021) and by NSF CHE-1900568 grant (A. B.).

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

## A    Finite-size analysis of the numerical data

In this Appendix we provide additional data on finite-size scaling and extrapolation procedure for the numerical data.

### A.1    Finite-size flow of the ratio $r$-statistics

First, we describe the procedure of the finite-size collapse of the ratio $r$-statistics. For each value of the bare hopping decay rate $a$ the ratio $r$-statistics has been calculated for the range of anisotropy parameters $\beta$ and system sizes $L$ (see Fig. 3(a) in the main text for $a = 1$ and Fig. 6(a, b, c) for $a = 0.5$, $a = 1.5$, and $a = 1.75$, respectively).

The first approximation of the transition $\beta = \beta_{AT}(a)$ is given by the crossing point of finite-size $r(\beta, N)$ curves. More accurate single-parameter collapse of all curves of the form

$$\langle r \rangle (\beta, L) = R(|\beta - \beta_{AT}|L^{1/\nu}) \tag{14}$$

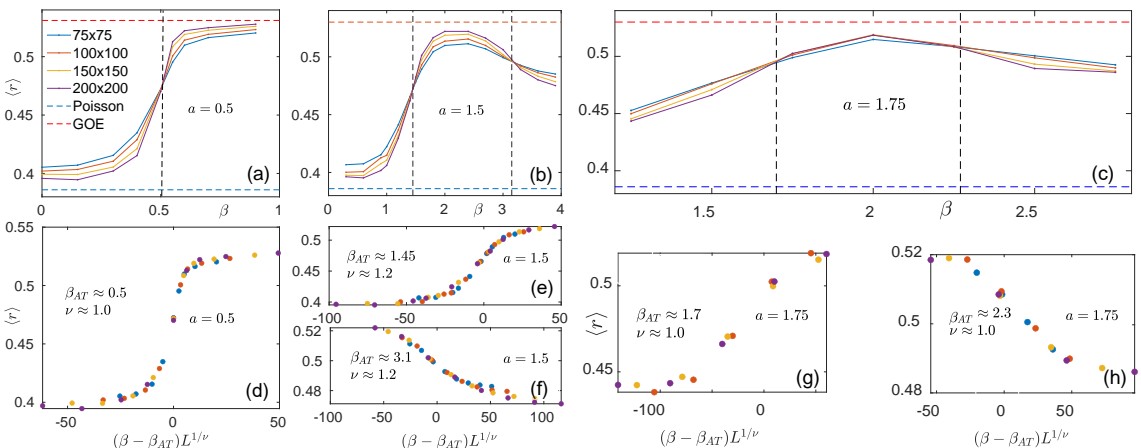

Figure 6: **Ratio $r$-statistics versus the anisotropy parameter** $\beta$ for the disorder strength $W = 10$, (a) $a = 0.5$, (b) $a = 1.5$, and (c) $a = 1.75$ for different system sizes $L = 75$, 100, 150, and 200 with the corresponding number of disorder realizations 2000, 2000, 1000, and 400, respectively. Dashed horizontal lines show the limiting ergodic ($r \simeq 0.53$) and Poisson ($r \simeq 0.386$) values. The vertical lines show the extracted anisotropy parameter $\beta_{AT}$ at the localization transition. Panels (d-h) show the finite-size collapse of all 5 crossing points with critical values $\beta_{AT}$ and critical exponents $\nu$.

provides best parameters $\beta_{AT}$ and $\nu$, see Fig. 3(b) and Fig. 6(d-h).

The black solid line in Fig. 1 in the main text shows the result for the critical value of $\beta_{AT}$ extracted from Fig. 3 for $a = 1$, which coincides with the analytical prediction, Eqs. (2), (3), within the $\sim 10$ %-errorbar. Figure 6(a-c) shows the similar data for extrapolation for $a = 0.5$, $a = 1.5$, and $a = 1.75$. The intersection points of $r$-statistics versus $\beta$ for different system sizes correspond to the Anderson localization transition and agree quite well with the analytical values shown in Fig. 1 of the main text. Figure 6(d-h) shows finite-size collapse which provides the critical $\beta_{AT}$ within 10 % error bar with respect to the analytically predicted values as well as the critical exponent $\nu \simeq 1$ close to the unity).

Second, in Fig. 7 we show the finite-size data of $r$-statistics versus the energy $E$ from which we have extrapolated the fraction $f_{\mathrm{erg}}$ of the ergodic states shown in the inset to Fig. 3(b) of the main text.

## A.2    Extrapolation of the multifractal spectrum $f(\alpha)$ and fractal dimensions $D_q$

In this subsection we provide the standard extrapolation procedure for the spectrum of fractal dimensions (see, e.g., [16, 17, 23, 34, 35]) and for the fractal dimensions $D_q$ [26].

For the former we use the following expression for the multifractal spectrum $f(\alpha, N)$ at finite system size $N = L^d$, $d = 2$

$$f(\alpha, N) = f(\alpha) + \frac{c_\alpha^{(1)}}{\ln N} + \frac{c_\alpha^{(2)}}{(\ln N)^2} + \dots \, , \tag{15}$$

with a certain $\alpha$-dependent constants $c_\alpha^{(k)}$. This follows from the definition of the multifractal spectrum $f(\alpha)$ given by the scaling of the probability distribution $\mathcal{P}(\ln |\psi_n(i)|^2) \sim N^{f(\alpha)-1}$

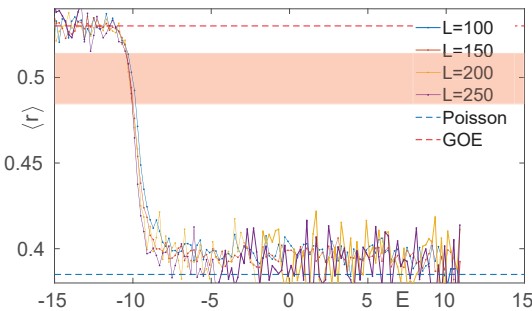

Figure 7: **Ratio $r$-statistics versus the energy** $E$ for the disorder strength $W = 10$, $a = 1.5$, and $\beta = 0.3$ for different system sizes $L = 100$, $150$, $200$, and $250$ with the corresponding number of disorder realizations 1000, 1000, 150, and 50, respectively. Dashed horizontal lines show the limiting ergodic ($r \simeq 0.53$) and Poisson ($r \simeq 0.386$) values. The red shaded region shows the threshold $r \in [0.48, 0.52]$ used for the extraction of the fraction $f_{\mathrm{erg}}$ of the ergodic states in the main text.

of the logarithm of the wave-function intensity $\alpha = -\ln |\psi_n(i)|^2 / \ln N$ [26] and extracted directly from the histogram over $\alpha$ [34–36]. Here and further we stick to the quadratic in $1/\ln N$ behavior in order to have reliable extrapolation.

The corresponding finite-size $f(\alpha, N)$ and extrapolated $f(\alpha)$ curves are given in Fig. 8 for a certain mid-spectrum energy $E = 5$ in the localized phase, $a = 1.5$, $\beta = -1$ and obey the normalization condition, $\max_\alpha f(\alpha) = f(\alpha_0) = 1$, of the probability distribution $\mathcal{P}(\alpha)$.

The position of the maximum $\alpha_0$ of $f(\alpha)$ and its slope $k = 1/\alpha_0$ corresponds to the effective power-law spatial decay of the wave function with the distance $r = |i - i_0|$ from its maximum $i = i_0$. Indeed, with the distance the eigenstate decays as $N^{-\alpha} = |\psi_n(i)|^2 \sim r^{-\gamma(a)}$, $\gamma(a) = 2\max(a, 2d - a)$, while the number of states increases as the volume $N^{f(\alpha)} \sim r^d$. Thus, resolving these expressions with respect to $r$ one obtains

$$f(\alpha) = \frac{\alpha}{\alpha_0}, \quad \alpha_0 = \frac{\gamma(a)}{d} = \max(a, 2d - a) \, , \tag{16}$$

which is confirmed by the numerical simulations, Fig. 8.

The finite-size fractal dimension is defined by the formula $D_q(N) = \ln I_q/(1 - q) \ln N$, with the generalized inverse participation ratio (IPR), $I_q = \left\langle \sum_i |\psi_n(i)|^{2q} \right\rangle = c_q N^{(1-q)D_q}$. Main contributions to it are given by the scaling exponent $D_q$ and the prefactor $c_q$ of IPR similarly to (15)

$$D_q(N) = D_q + \frac{(1 - q)^{-1} \ln c_q}{\ln N} \, . \tag{17}$$

The resulting extrapolated $D_2$ is shown in Fig. 9 versus $a$ for $\beta = 2$. One can see there (yellow squares) the transition from localized phase $a > 2$ with $D_2 \to 0$ to the extended one, $D_2 > 0$, at $a < 2$. As a reference point we show the fractal dimension for the power-law random banded matrix (PLRBM) model [26] extrapolated using the simple linear formula (17). The discrepancy between these models in the extended phase is due to severe finite-size effects in anisotropic model (we address this issue in the next Appendix).

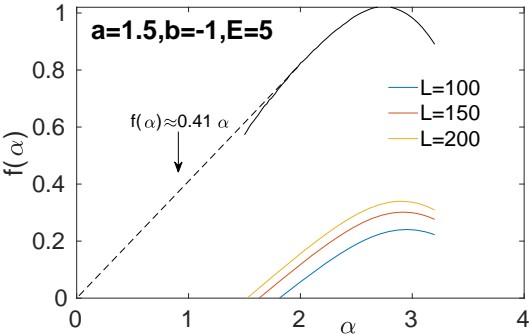

Figure 8: **Finite-size extrapolation of the multifractal spectrum** $f(\alpha)$ for the energy $E = 5$, disorder strength $W = 10$, $a = 1.5$, and $\beta = -1$. $f(\alpha)$ is extrapolated from $L = 100$, 150, and 200 with the corresponding number of disorder realizations 1000, 500, and 100, respectively.

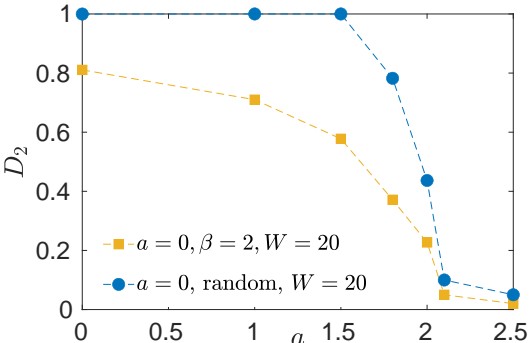

Figure 9: **Comparison of extrapolated $D_2$ versus** $a$ for the anisotropic model with fixed bare disorder $W = 10$ (yellow squares) and for the $2d$ power-law random banded model (blue circles). The anisotropy is taken to be $\beta = 2$. $D_2$ are extrapolated from $L = 100$, 150, 200, and 250 with the corresponding number of disorder realizations 1000, 500, 100, and 50, respectively.

### A.3   Inverse participation ratio and the fraction of ergodic states

Here we focus on the estimation of the fraction of ergodic high-energy states in the localized state at $0 < \beta < a < 2$. In order to check Eq. (8) of the main text and results extracted from Fig. 7 we consider the plot of energy-dependent IPR values sorted in increasing order for different system sizes versus the renormalized fraction of the states $(n/L)^{3-a}/\ln L$, see Fig. 10. Panels (a) and (b) show the IPR itself $I_q$ and its renormalization $N \cdot I_q$ in order to emphasize the scaling of the localized and ergodic states, respectively, given as an inset to Fig. 3(b) in the main text. The same analysis has been done in Fig. 7 for the ratio $r$-statistics versus energy.

## B   Numerical evidence of the localization at $a < d$

In this section we provide the numerical evidence of the localization of the measure 1 of the states in the localized phase at $\beta < a < d$ and $a > d$. First, we consider static properties of the

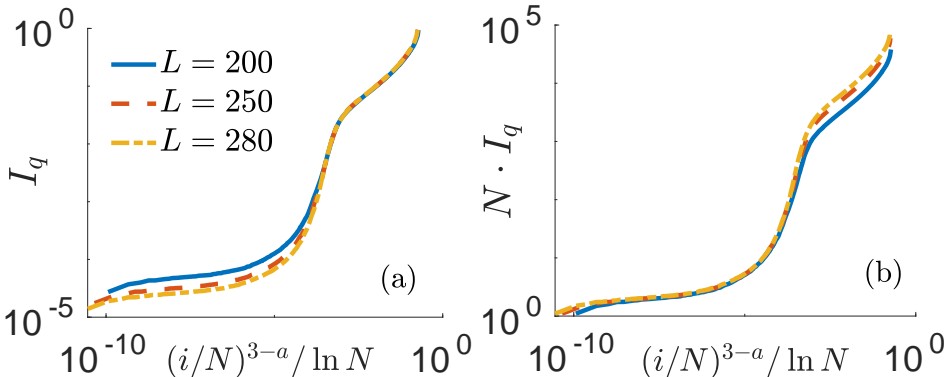

Figure 10: **Inverse participation ratio sorted in increasing order versus renormalized state index** (a) IPR itself showing collapse at the localized states and (b) IPR renormalized to the system size $N$ showing the collapse for ergodic states. The disorder strength for $a = 1$, $\beta = 0.3$ is $W = 20$. Finite size data is represented for $L = 200$ (solid blue), 250 (dashed red), and 280 (dash-dotted yellow) with the corresponding number of disorder realizations 100, 80, and 50, respectively.

wave function spatial decay. And second, we provide the dynamical measure of localization via the return probability.

## B.1  Wavefunction spatial decay

Similar to Figs. 4(b) and 5(b) in the main text and the results of [16–18], we consider the typical wave function spatial decay with the distance with respect to its maximum. Fig. 11 confirms the duality of power-law spatial decay rate $\gamma(a) \approx \gamma(2d - a)$ [16–18] in the localized phase of the anisotropic model between the standard locator expansion states $a > d = 2$ and beyond it $a < d$.

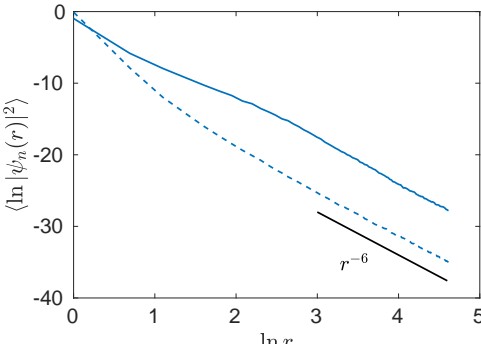

Figure 11: **Power-law spatial decay of eigenstates** in the bulk of the spectrum for $a = 1$ (solid), 3 (dashed), $\beta = 0.5$, at the system size $L = 200$ with 200 disorder realizations. The disorder amplitude is taken to be $W = 20$ for $a = 1$ and $W = 200$ for $a = 3$ in order to make the power-law tail dominant on moderate sizes in both cases.

## B.2   Wavepacket dynamics. Return probability

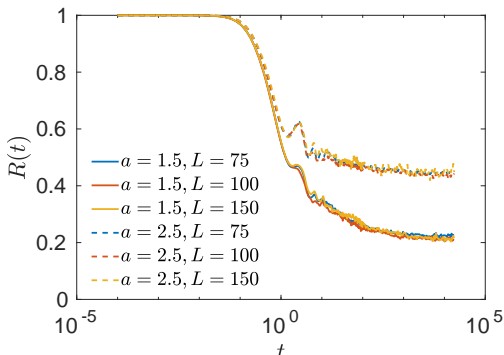

Figure 12: **Survival probability of the delta-peak initialized wavepacket in the localized phase** for the disorder strength $W = 20$, $\beta = 0.3$, and $a = 1.5$ (solid lines), $a = 2.5$ (dashed lines) for different system sizes $L = 75$, 100, and 150 with the corresponding number of disorder realizations 1000, 750, and 400, respectively.

For this purpose we initialize the wavepacket with the delta function at time $t = 0$

$$\psi(0) = \delta(x - x_0) = \sum_n \psi_n^*(x_0, 0)\psi_n(x, 0) \tag{18}$$

and compute the evolution of it in time

$$\psi(t) = \delta(x - x_0) = \sum_n \psi_n^*(x_0, 0)\psi_n(x, t) = \sum_n \psi_n^*(x_0, 0)\psi_n(x, 0)e^{-E_n t} \tag{19}$$

by considering the survival probability defined as [60–62]

$$R(t) = |\langle\psi(0)|\psi(t)\rangle|^2 = \sum_{n,m} |\psi_n(x_0, 0)|^2 |\psi_m(x_0, 0)|^2 e^{-(E_n - E_m)t} \ . \tag{20}$$

which is an important dynamical measure relevant also for many-body localization [63, 64].

By definition at time $t = 0$ the survival probability equals unity and then as time evolves it decays (with some revivals) to the constant value at long times

$$R(t \to \infty) = \sum_n |\psi_n(x_0, 0)|^4 \ , \tag{21}$$

analogously to the IPR with the summation over energies, but not coordinates. The scaling of the latter measure with the system size $N$ shows the localization properties of the wavepacket and, thus, of the underlying eigenstates.

Figure 12 shows the survival probability at $\beta = 0.3$, $a = 1.5$ and $a = 2.5$, corresponding to the localized phase $\beta < a < d = 2$ and $a > d$, respectively, for several system sizes, averaged over the disorder realizations and several initial coordinated $x_0$. From the data it is clearly seen that in both cases the limiting value (21) does not scale with the system size confirming the localization of the eigenstates. The larger limiting value of $R(t)$ for $a > d$ corresponds to the smaller localization region of localized states with respect to $\beta < a < d$ according to the renormalization group predictions.

## C   Numerical characterization of extended phase

In this Appendix we characterize the extended phase of the considered anisotropic model using more quantities from the multifractal analysis.

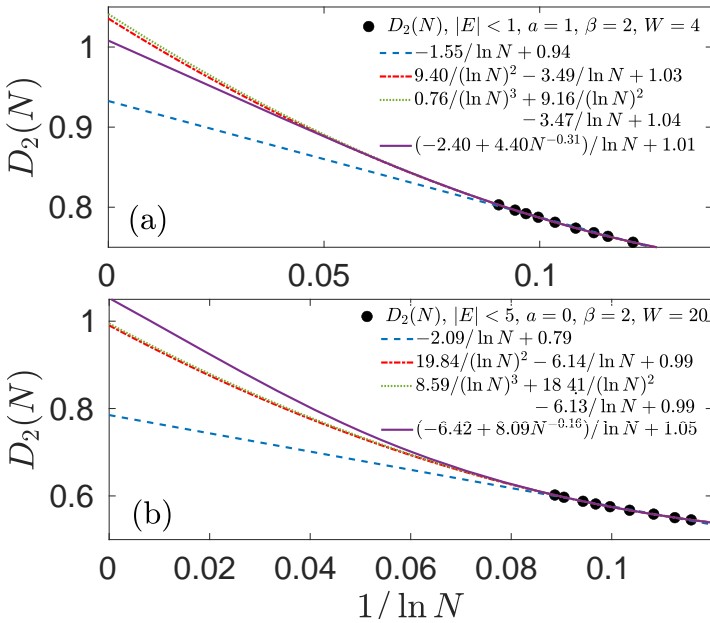

Figure 13: **Finite-size extrapolation of the fractal dimension** $D_2$ (symbols) with linear (blue dashed), quadratic (red dash-dotted), cubic (green dotted) expressions in $x = 1/\ln N$ as well as the one with irrelevant exponent (violet solid) considered in [30]. We show two parameter sets (upper panel) $a = 1$, $\beta = 2$ is $W = 4$ and (lower panel) $a = 0$, $\beta = 2$ is $W = 20$ in order to emphasize that this issue present both at weak and strong disorder. $D_2$ is averaged over the energy interval $|E| < W/4$ and extrapolated from $L = 75, 85, 100, 125, 150, 175, 200,$ and 250 with the corresponding number of disorder realizations 2000, 2000, 2000, 2000, 1000, 600, 600, and 300, respectively.

First, we should mention that the extrapolation of $D_2$ in this case is more subtle. Due to limited system sizes in 2d the linear approximation (17) provides unreasonable results and, thus, following recent literature we use quadratic in $1/\ln N$ extrapolation and compare it with further cubic one both for weak and strong disorder, see Fig. 13 [65]. In order to double check we also fit the data with the expression with irrelevant exponent suggested in [30]

$$D_q(N) = D_q + \frac{(1-q)^{-1} \ln c_q + \gamma_q N^{-y_{irr}}}{\ln N} \ .$$ (22)

All the results confirm the ergodic nature of the extended phase in the considered model [66] which is spoiled by severe finite-size effects forcing one to go beyond linear extrapolation, Eq. (17).

### C.1   Finite-size behavior of the wave function decay

The wave function spatial decay at several system sizes shows that in the extended phase the power-law decay with dual decay rate $\gamma(a) = 2 \max(a, 2d - a)$ develops only at small distances

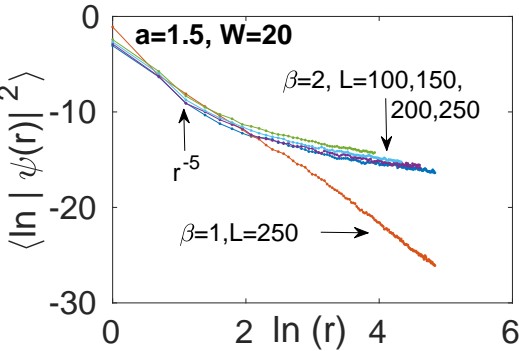

Figure 14: **Spatial wave function decay in the extended phase, $a = 1.5$, $\beta = 2$, for different system sizes.** in comparison to the one in the localized phase (red line), $a = 1.5$, $\beta = 1$. The disorder strength is taken to be $W = 20$. System sizes are $L = 100$, $150$, $200$, and $250$ with the corresponding number of disorder realizations 1000, 500, 100, and 50, respectively.

$|i - i_0| \equiv r < r_0 \sim L^m$ which increases with the system size slower than the length scale, $m < 1$, Fig. 14 At larger distances the power-law tail with smaller decay rate $c \leq d$ grows

$$|\psi_n(i)|^2 \sim \begin{cases} L^{-A} \left(\frac{r}{r_0}\right)^{-\gamma(a)}, & r < r_0 \\ L^{-A} \left(\frac{r}{r_0}\right)^{-c}, & r > r_0 \end{cases} \tag{23}$$

and eventually it leads to the decay of the wave function maximum $|\psi_n(i_0)|^2 \sim L^{\gamma m - A}$, which is consistent with the ergodic typical value $|\psi_n(L)|^2 \sim L^{-A-(1-m)c} \sim L^{-d}$, as $A = d - (1-m)c$, confirming, thus, Fig. 4(a) of the main text. In the Fig. 14 we have $c \simeq 1$, $A - \gamma m \simeq 0.54$, with $m \simeq 0.12$.

## C.2 Higher-order level statistics

Another interesting measure is the higher order of the ratio $r$-statistics [67, 68] generalizing the standard one [28, 29]. Indeed, the probability distribution $P(s, n)$ of the level spacing $s_{k,n} = (E_{k+n} - E_k)/\delta$ of the $n$-consecutive energies separated by $n - 1$ levels in between, renormalized by the mean level spacing $\delta = \langle E_{k+1} - E_k \rangle$, both in the Wigner-Dyson and Poisson limit gives the same mean value $s = n$ while its variance determines the level rigidity and scales as $\sim \ln n$ ($\sim n$) for Wigner-Dyson (Poisson) case.

The analysis of the above distribution $P(s, n)$ in the extended phases of PLRBM and considered anisotropic model shows the Wigner-Dyson behavior of the former level rigidity and the Poisson scaling of the variance for the anisotropic non-random hopping case, Fig. 15(a-b).

The corresponding $r_n$-statistics,

$$r_n = \left\langle \frac{\min(s_{k,n}, s_{k+1,n})}{\max(s_{k,n}, s_{k+1,n})} \right\rangle \tag{24}$$

of the considered anisotropic model shows Wigner-Dyson behavior at small $n \lesssim 5$ (cf. solid lines and the dashed one in Fig. 15(c)) and values consistent with Poisson at large $n \gtrsim 60$ after rescaling $n L^{0.8}$.

## C.3 Overlap correlation function $K(\omega)$ and the correspondence to fractal dimensions

The overlap correlation function

$$K(\omega = E_n - E_{n'}) = N \sum_i \langle |\psi_n(i)|^2 \, |\psi_{n'}(i)|^2 \rangle \tag{25}$$

is an important measure of the wave function statistics [69, 70]. It is the Fourier-transform of the survival (or return) probability [60–62] which is an important dynamical measure relevant also for many-body localization [63, 64].

The power-law decay rate $1 - D_s$ of $K(\omega)$ at small $\omega$ is usually related to the fractal dimension $D_2$ [71–73]

$$D_s = D_2 \ , \tag{26}$$

however, in later works [35, 60, 74] this statement was generalized to the following: if the position of the crossover between two different power-law decays at small and large frequencies (associated with the miniband size [35, 60]) does not scale with $N$, Eq. (26) is valid [74].

In Fig. 16 we show the typical plot of $K(\omega)$ in the extended phase of the considered model. Following the Chalker scaling, Eq. (26), $D_s$ is shown to be close to $D_2$, while the decay rate $\kappa_2$ at larger $\omega$ approaches the dimensionality $\kappa_2 \to d = 2$ with increasing effective disorder according to [74].

# D   Spectrum of hopping, Eq. (5)

The spectrum of the hopping term $V_{ij} = -\frac{1-\beta\cos^2\phi_{ij}}{r_{ij}^a}$ from Eq. (1) is given by its Fourier transform due to translation-invariance of hopping

$$V_q = -\sum_{i,j} e^{iq_x(i_x - j_x) + iq_y(i_y - j_y)} \frac{1 - \beta\cos^2\phi_{ij}}{r_{ij}^a} \ . \tag{27}$$

For $a \neq d = 2$ the latter can be calculated in the continuous approximation as

$$V_q = -\int_0^\infty r dr \int_0^{2\pi} d\phi e^{iqr\cos(\phi - \phi_q)} \frac{1 - \beta\cos^2\phi}{r^a} = c_a q^{a-2} \left[ \beta - a - (2 - a)\beta\cos^2\phi_q \right] \ . \tag{28}$$

Here $c_a = \pi 2^{1-a} \frac{-\Gamma(-a/2)}{\Gamma(a/2)}$, $\Gamma(a)$ is a Gamma-function, and $q = \pi n/L$ is the quantized momentum ,with integer $n \lesssim L/a_0$, $a_0$ is the inter-atomic distance which we choose to be unity $a_0 \equiv 1$ without loss of generality. The special case of $a = d = 2$ should be considered separately as the result depends explicitly on $a_0$

$$V_q = \pi \left[ (2 - \beta)\left(\gamma_E + \ln\left(qa_0/2\right)\right) - \frac{\beta}{2}\cos(2\phi_q) \right] \ , \tag{29}$$

with the Euler – Mascheroni constant $\gamma_E \simeq 0.577216$.

The divergence of both Eqs. (28) and (29) at $q \to 0$ at $a \leq d$ signals on the presence of (the measure zero of) high-energy delocalized states [17, 18].

# E  Main idea of the renormalization group analysis

In this Appendix we follow [15, 18] and reproduce the idea of the renormalization group (RG) analysis for the 2d anisotropic system. Similarly to [15] let's take the disorder amplitude $W \gg 1$ to be large compared to the nearest-neighbor hopping $V_{i,i+1}$ and apply the RG procedure to study this problem. As a step of the RG we first cut off the tunneling at a certain scale $R_0$ and calculate the wavefunctions ($R_0$ modes) for this scale. Then new cutoff $R_1 \gg R_0$ is chosen and new modes ($R_1$ modes) are constructed as a superposition of $R_0$ modes. The localization length increases from $\ell_0 \lesssim R_0$ to $\ell_1 \lesssim R_1$ due to the presence of resonances. Due to the presence of large parameter $W \gg 1$ only pairs of resonances are taken into account (please see [18] for more details). The annihilation operators $\widehat{\psi}_k^{(1)}$ of new $R_1$ modes can be written via the initial site annihilation operators $\widehat{c}_m$ as follows

$$\widehat{\psi}_k^{(1)} = \sum_i \psi_k^{(1)}(i)\widehat{c}_i \ . \tag{30}$$

Thus, the hopping term $V_{ij} = -\frac{1 - \beta \cos^2 \phi_{ij}}{r_{ij}^a}$ rewritten in new operators takes the form

$$\sum_{i,j} V_{ij}\widehat{c}_i^\dagger \widehat{c}_j = \sum_{k,l} \widehat{\psi}_k^{(1)\dagger}\widehat{\psi}_l^{(1)} \sum_{i,j} \psi_k^{(1)}(i)\psi_l^{(1)*}(j)V_{ij} \ . \tag{31}$$

According to RG assumption the modes $\psi_k^{(1)}(m)$ are localized $r_{km} < \ell_1$ at the length $\ell_1 \lesssim R_1$, thus, one can neglect the difference between $V_{ij}$ and $V_{kl}$ ($|r_{ij} - r_{kl}| < r_{ik} + r_{jl} < 2l_1 \lesssim R_1$). As a result, Eq. (31) reads as

$$\sum_{i,j} \frac{\widehat{c}_i^\dagger \widehat{c}_j}{r_{ij}^a} \simeq \sum_{k,l} \frac{t_0 l_k l_l^*}{r_{kl}^a}\widehat{\psi}_k^{(1)\dagger}\widehat{\psi}_l^{(1)} \ , \tag{32}$$

with the effective charge $l_k = \sum_i \psi_k^{(1)}(i)$.

In order to estimate the renormalized hopping term $l_k l_l^* / r_{kl}^a$ let's consider the mean squared value of $l_k$ at a certain energy $E$ as follows

$$\langle l^2 \rangle_E = \frac{\langle \sum_k l_k^2 \delta(E - E_k) \rangle}{\rho(E)} =$$

$$\frac{\left\langle \sum_k \sum_{i, r_{ik} < R_1} \sum_{j, r_{jk} < R_1} \psi_k^{(1)}(i)\psi_k^{(1)*}(j)\delta(E - E_k) \right\rangle}{\rho(E)} \simeq$$

$$\frac{\sum_{r_{ij} < R_1} \langle \operatorname{Im} G_{i-j} \rangle}{\rho(E)} \simeq \frac{\operatorname{Im} \bar{G}_{q \simeq 1/R_1}(E)}{\rho(E)}, \tag{33}$$

Here the density of states (DOS) is given by

$$\rho(E) = \left\langle \sum_k \delta(E - E_k) \right\rangle = \frac{1}{N}\sum_q \operatorname{Im} \bar{G}_q(E) \ . \tag{34}$$

Taking into account that the imaginary part of the Green's function is given by a Lorenzian

$$\operatorname{Im} \bar{G}_q(E) \simeq \frac{W}{(E - V_q)^2 + \pi W^2/12} \ , \tag{35}$$

in the coherent potential approximation, one can straightforwardly finds that the DOS is $q$-independent and is determined solely by the disorder amplitude (like in [18])

$$\rho(E) \simeq \int \frac{d^d q}{(2\pi)^d} \frac{W}{(E - t_0 q^{a-d})^2 + W^2} \simeq \frac{1}{W} \ . \tag{36}$$

Here we consider for simplicity the box distribution of the disorder $-W/2 < \mu_i < W/2$ with the finite variance $\langle \mu_i^2 \rangle = W^2/12$ and use it in the determination of the self-energy of the Green's function.

As a result

$$\operatorname{Im} \bar{G}_{q \sim 1/R_1}(E) \simeq \frac{W}{R_1^{2(d-a)}} \ , \tag{37}$$

and the effective hopping within the RG approximation scales as

$$V_R^{eff} = \min \left( \frac{t_0}{R^a}, \frac{W^2}{t_0 R^{2d-a}} \right) \ . \tag{38}$$

giving localization with the characteristic change of the power law tail at $R \simeq W^{1/(d-a)} \gg 1$.

Eventually in the case $W \gg 1$ this estimate provides the localization of all eigenstates at $E \lesssim W \sim O(1)$ and the duality of the polynomial decay rate of the corresponding wave functions, $a_{eff} = \max(a, 2d - a)$ at $a < d$ and $a > d$. For more rigorous consideration of RG procedure, please see [15, 18].

# F Ioffe-Regel criterion

In this Appendix we estimate the energy-dependent mean-free path for $a < d$ and apply the Ioffe-Regel criterion of localization in order to estimate the fraction of ergodic states in the localized phase of the considered anisotropic model.

The mean-free path at a certain energy $E$ can be estimated as follows

$$\ell_{mfp}(E) \simeq v_{q_E} \tau_{q_E} \ , \tag{39}$$

where $q_E$ and $v_q$ are determined from the following equations

$$V_{q_E \lesssim 1} = E, \Rightarrow q_E \sim \min \left[ 1, E^{-1/(d-a)} \right] \tag{40}$$

$$v_q = \frac{dV_q}{dq} \sim q^{a-d-1} \ , \tag{41}$$

while the level broadening can be estimated with Fermi Golden rule of the scattering of plane waves on the impurities $\mu_i \sim W$

$$\tau_{q_E}^{-1} = \operatorname{Im} G_{i-j=0}(E) \simeq \rho(E) \frac{W^2}{12} \ . \tag{42}$$

Small $q_E$ corresponds to large energies $E \gg W$, thus, the DOS at such energies is not anymore determined by (36), but involves $q_E$ as follows

$$\rho(E \gg W) = \frac{d^d q_E}{dV_{q_E}} \sim q_E^{2d-a} \ . \tag{43}$$

As a result using (36) we obtain

$$\ell_{mfp}(E) \sim W^{-2} q_E^{2a-3d-1} \sim W^{-2} E^{(3d+1-2a)/(d-a)} \ . \tag{44}$$

According to the Ioffe-Regel criterion the states are delocalized

- in $d = 1$ as soon as $\ell_{mfp} > L$;

- in $d = 2$ as soon as $\ell_{loc} \sim e^{cq_E \ell_{mfp}} > L$;

- in $d = 3$ as soon as $q_E \ell_{mfp} > 1$.

leading to a certain upper cutoff $q_E < q_*$. The fraction of such delocalized states is given by

$$f_{\mathrm{erg}} = \int_0^{q_*} d^d q \sim q_*^d \ . \tag{45}$$

After straightforward algebra the mobility edge can be estimated as

- in $d = 1$

$$q_E < q_* = \left(W^2 N\right)^{-\frac{1}{2(2-a)}} \Rightarrow f_{\mathrm{erg}} \sim q_* \sim N^{-\frac{1}{2(2-a)}}; \tag{46}$$

- in $d = 2$

$$q_E < q_* = \left(W^2 \ln N\right)^{-\frac{1}{2(3-a)}} \Rightarrow f_{\mathrm{erg}} \sim q_*^2 \sim \ln N^{-\frac{1}{3-a}}; \tag{47}$$

- in $d = 3$

$$q_E < q_* = W^{-\frac{2}{9-2a}} \Rightarrow f_{\mathrm{erg}} \sim q_*^3 \sim O(1) \ . \tag{48}$$

In the 2d case considered in the main text (see the inset to Fig. 3(b)) the fraction of ergodic states decays as the power of the logarithm of $N$.

Note that following [17, 75] one can find the fraction of modes which are localized in the momentum $q$-basis. This condition is related to the level spacing $|V_{q_E} - V_{q_E+\pi/L}|$ to be of the order of the corresponding hopping

$$|V_{q_E} - V_{q_E+\pi/L}| \sim \frac{v_{q_E}}{N^{1/d}} > \frac{W}{N^{1/2}} \tag{49}$$

which leads to

- in $d = 1$

$$q_E < q^{**} = \left(\frac{N^{1/2}W}{t_0}\right)^{-\frac{1}{(2-a)}} \simeq q^* \tag{50}$$

- in $d = 2$

$$q_E < q^{**} = \left(\frac{W}{t_0}\right)^{-\frac{1}{3-a}} \ll q^* \tag{51}$$

- in $d = 3$

$$q_E < q^{**} = \left(\frac{W}{N^{1/6}t_0}\right)^{-\frac{1}{4-a}} \ll q^* \ . \tag{52}$$

Note that the localization in the momentum $q$-basis is more restrictive for all $d > 1$ as it provides the fraction of plane wave modes, while most of delocalized modes in $d \geq 2$ are of diffusive nature.

# G  Related models and feasible experimental setup

Some similar models based on Eq. (1) are also considered numerically: for instance, the hopping model $t_{ij} = \cos(kr_{ij})/r_{ij}^a$ (see, e.g., [76, 77] and the hopping model $t_{ij} = (1 + \eta_{ij})/r_{ij}^a$ [16, 18, 23] with uncorrelated random $\eta_{ij}$. Unlike the considered model, Eq. (1), these two models show only ergodic delocalized states when $a < d = 2$. However, extended states for $\beta > a$ with Wigner-Dyson statistics are also observed in the power-law Euclidean random matrix models when $a < d = 2$ (The details of calculation will be shown in further publications). Such matrix ensemble is generated from the uniform-random-distributed quantum dipoles in a square lattice. As an experimentally feasible setup one can consider a set of ions trapped in individual microtraps, which allows for arbitrary geometries and easy control over the effective anharmonicity of the spatial ion motion near the microtrap minima. Spin-dependent optical dipole forces applied to such ionic crystal create long-range effective spin-spin interactions and allow the simulation of spin Hamiltonians that possess nontrivial phases and dynamics. By tailor the optical forces one can generate arbitrary interactions between spins. Our findings could be observed in the flip-flop spin-model, as well as in the phonon hopping model itself.

Another way to realize long-range anisotropic model would be to use the dipole radiation in a 2d photonic crystal near the Dirac cone (i.e., dipolar interaction mediated by the photonic Dirac cone between atoms), see Ref. [78] in which the authors obtain effective long-range interactions $1/r^{1/2}$, based on the results of Ref. [79]. The $1/r$ hopping $(\delta_{ab} - n_a n_b)/r$ can be as well relevant for 2d polaritons [80].

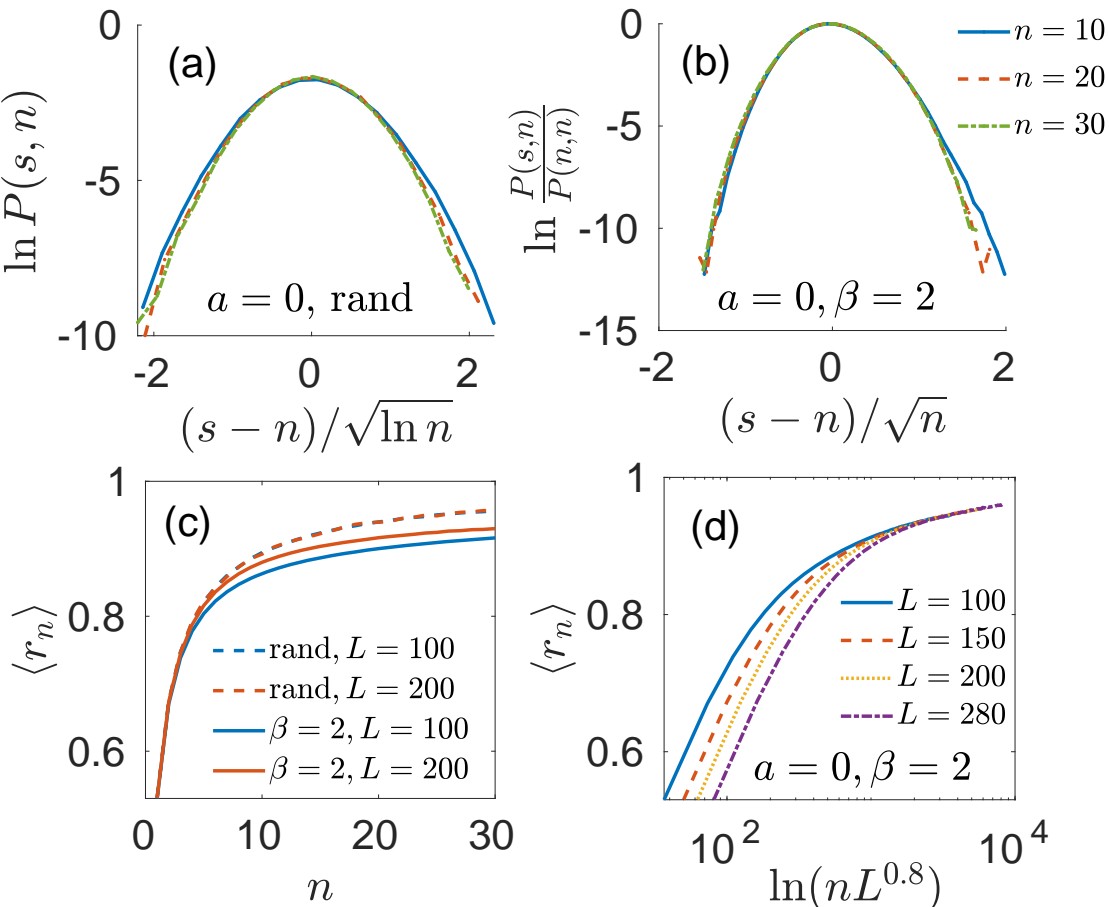

Figure 15: **Higher-order level spacing distribution $P(s,n)$ and ratio level statistics $r_n$.** Upper row shows the collapse of $P(s,n)$ for different $n$ (shown in legend) for (a) the $2d$ power-law random banded model showing Wigner-Dyson variance $\sim \ln n$, and (b) the anisotropic model showing Poisson level variance $\sim n$ at large $n \gtrsim 10$. Lower row shows the collapse of ratio $r$-statistics for different system sizes $L$ (shown in the legend) (c) at small $n$ (in the whole interval) for the anisotropic model [solid] (PLRBM [dashed]), and (d) at large $n \gtrsim 60$ for the anisotropic model by the rescaling of $n$ to $nL^{0.8}$ In both models $a = 0$ and $W = 20$. In the anisotropic model $\beta = 2$. The number of disorder realizations is 1000, 500, 100, and 50 for $L = 100$, 150, 200, and 280, respectively.

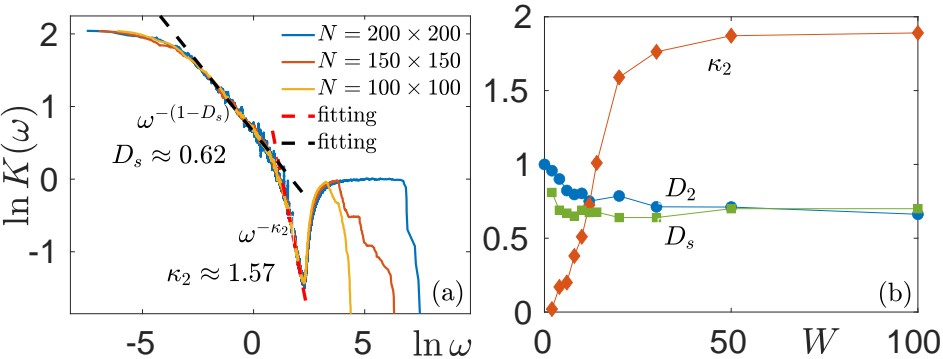

Figure 16: **Overlap function $K(\omega)$ and the comparison of its decay rates to fractal dimensions $D_2$.** (a) $K(\omega)$ for different system sizes $L$ (shown in legend) in log-log scale. The power-law fitting gives $K(\omega) \sim \omega^{-\kappa}$, with $\kappa = 1 - D_s$ ($\kappa_2$) for small (large) $\omega$. The parameters are $a = 0$, $\beta = 2$, and $W = 20$. (b) the comparison of the decay rates $1 - D_s$ and $\kappa_2$ of $K(\omega)$ with the fractal dimension $D_2$ versus the disorder strength $W$ corresponds to the Chalker scaling $D_2 = D_s$. $D_2$, $D_s$, and $\kappa_2$ are extrapolated from $L = 100$, $150$, $200$, and $250$ with the corresponding number of disorder realizations $1000$, $500$, $100$, and $50$, respectively.