# Peer review of "Anisotropy-mediated reentrant localization"

_SciPost Physics_

## Round 2 · Referee Report · Anonymous (Referee 1) · 2021-9-25

Strengths

1 - 2D Anderson transition for the effective long-range tunneling model coming from diluted tilted dipoles is studied apparently showing an interesting reentrant transition
2 - Extensive numerical data support the analysis
3 - The numerics is supported by renormalization group analysis

Weaknesses

1 - The presentation style and the construction of the paper makes it difficult to follow - see report
2 - Some of numerical results are poorly described with insufficient details so the reader may find it difficult to reproduce the data
3 - Are all the data shown necessary to reach the conclusions of the paper?
4 - Interpretation of the numerics seems controversial

Report

Scientifically the paper seems, at first glance, sound and gives a comprehensive analysis of Anderson localization and spectral properties in an interesting and physically realisable model. The style of the paper makes it, however, extremely difficult to read and understand. Section 2 is entitled "Models and methods", section 3 "methods". What is the difference between methods in 2 and 3? The main body of the paper is rather short and most of the results are hidden in Appendixes which makes the reader jump back and forth unnecessarily. For example Fig.4 in the main text shows the spectrum of fractal dimensions, $f(\alpha)$, (p.9) which is not defined in the main text but is mentioned in A.2 pp. 18-19. Similarly fractal dimension $D_2$ illustrates multifractality in Fig.2 (p.5) with definition again hidden in Appendixes. In the present form the manuscript is unsuitable for publication in Scipost mainly because of the form of the presentation. Also the scientific content which seemingly is sufficiently high requires reconsideration, see requested changes and the comments below.

Let me exemplify the problem with the analysis of level spacing ratio (gap ratio) statistics. For systems with energy dependent mobility edge the safe, standard approach is to find the mean ratio, say $r$, in narrow energy intervals (see e.g. Luitz et al. 2015, PRB) averaging over different disorder realisations. It seems to be indeed done, somehow, in Fig.2b) where $r$ is plotted as a function of energy. While for $\beta=1,2$ $r$ seems not to be dependent on energy and shows signs of localized and extended phases, respectively, the intermediate $\beta=1.5$ case seems to show significant energy dependence. While in the caption we can read that data are for $L=250$, one cannot find out the width of energy window used for this plot (each yellow point for $\langle r\rangle$ corresponds, I believe, to some tiny energy interval). It is written that vertical dashed lines provide "the position of FSME extracted from finite-size data for r-statistics". But in fact the dashed lines point to the edge of the spectrum where, at low energies, density of states (DOS a) panel) significantly drops indicating the edge of the system. This cannot be the position of the mobility edge!.
It is not written in the caption which "r-statistics" is being considered. If this is the one shown in Fig.3 then it is performed badly. As written, energies from $[-W/2,W/2]$ interval, i.e. "95% of all the states" are taken for analysis. But this is simply wrong, as mean $r$ depends on energy, see Fig.2b) - the authors are again referred to e.g. Luitz et al. seminal paper.
The proper way of doing the analysis, let me repeat, would be to devide the energy interval $[-W/2,W/2]$ in which the bulk of the spectrum is contained into say 20 or 30 intervals and make the finite size analysis ala Fig.3 in each of the interval separately getting the critical $\beta(E)$ value depending on energy.
Authors are aware of the strong energy dependence of eigenstates, in particular in Fig.8 they study $f(\alpha)$ around energy $E=5$.
By the way, a similar analysis could be done for different disorder amplitude $W$ values as the manuscript considers a single choice $W=20$.
Panel 2c) shows the dimension $D_2$ which is both strongly energy dependent for $\beta=1.5$ (again - how yellow curve is obtained?). Its noninteger value suggests a fractal character. Multifractality requires, however, that $D_q$ change with $q$. The statement in the caption that panel b) and c) "show localized, multifractal and ergodic eigenstate properties in the spectral bulk" is thus wrong. In particular panel b) shows mean $r$ which is the eigenvalues property and is not linked with multifractality.
Appendices contain a number of results that are not used or referred to in the main text. I believe a selection should be made validating the claims of the paper (or showing its limitations). They are described sometimes not precisely. While, as mentioned above $f(\alpha)$ is analysed around $ E=5$ what about the following $D_q$ analysis? In Fig.9 a comparison with power-law random banded matrices is shown but is the power law assumed similar to that of the dipole model (1)? In the caption we read about 2d power-law random banded model - how 2d model is constructed? The referee (and [26] cited) knows only a standard PLRBM model where no dimensional structure is present. The precise definition of PLRBM used should be given in 2 lines explaining e.g. what is the sense of $W$ in PLRBM? Is energy dependence included in the analysis of IPR in Fig.10? Is energy dependence included in the higher order gap ratio statistics that appears in Sec.C of the Appendix and fig.15? Let me stop here asking the questions about the wealth of the material scattered throughout appendices.

Requested changes

1- I strongly recommend rewriting the paper is a style suitable for Scipost Physics. All notions and observables used in the main text must be properly defined before their use in the text and/or figures. More detailed changes: a/In the caption of Fig.1 it is written that the transition occurs at $r\simeq 0.47$ with $r$ defined in Eq.(4) while in Fig.2 or Fig.3 $\langle r\rangle$ is used to describe the vertical axis; please unify the notation. Strictly the level statistics is given as a set of $r_{n,1}$ values with the average $r$. This should be stated around (4). b/it is not frequent to call an earlier paper of one of coauthors as a pioneering paper'' (p.4). Maybe the use ofearly'' instead of ``pioneering'' would sound better? c/The level statistics ratio analysis should be performed in energy dependent style as described in the report. d/Similar comments affect other measures used in the paper/appendix. e/Authors should limit the numerical results presented in the appendices to a subset needed for the confirmation of the work claims and its conclusions. f/For example what is the relevance of the "spectrum of hopping" (appendix D) as well as Ioffe-Regel criterion (F) for the content of the paper? Either discuss this issues and their relevance to the main body of the paper or remove them for clarity.

  • validity: poor
  • significance: good
  • originality: good
  • clarity: low
  • formatting: mediocre
  • grammar: excellent

Author:  Ivan Khaymovich  on 2022-06-29  [id 2619]

(in reply to Report 1 on 2021-09-25)

We are very grateful to the Referee 1 for her/his very detailed review of our paper. Below we present the list of amendments resulting from the comments of the Referee.

Strengths 1 - 2D Anderson transition for the effective long-range tunneling model coming from diluted tilted dipoles is studied apparently showing an interesting reentrant transition 2 - Extensive numerical data support the analysis 3 - The numerics is supported by renormalization group analysis

Report: Scientifically the paper seems, at first glance, sound and gives a comprehensive analysis of Anderson localization and spectral properties in an interesting and physically realisable model.

Reply: We thank the Referee 1 for high evaluation of our manuscript both model-, numerical-, and analytical-wise.

Weaknesses 1 - The presentation style and the construction of the paper makes it difficult to follow - see report

Report: The style of the paper makes it, however, extremely difficult to read and understand. Section 2 is entitled "Models and methods", section 3 "methods". What is the difference between methods in 2 and 3? The main body of the paper is rather short and most of the results are hidden in Appendixes which makes the reader jump back and forth unnecessarily. For example Fig.4 in the main text shows the spectrum of fractal dimensions, f(α)f(α), (p.9) which is not defined in the main text but is mentioned in A.2 pp. 18-19. Similarly fractal dimension D2D2 illustrates multifractality in Fig.2 (p.5) with definition again hidden in Appendixes. In the present form the manuscript is unsuitable for publication in Scipost mainly because of the form of the presentation.

Requested changes 1- I strongly recommend rewriting the paper is a style suitable for Scipost Physics. All notions and observables used in the main text must be properly defined before their use in the text and/or figures.

Reply: We thank the Referee 1 for pointing out this important issue of presentation. We agree that the structure and even the names of the sections were not appropriate. In the revised version of the manuscript, we have restructured the main text by implementing most of appendices into it and by making a clear and logical construction of the text (please see the list of changes in the revised version and the revised version of the manuscript with highlighted changes attached).

Weaknesses 2 - Some of numerical results are poorly described with insufficient details so the reader may find it difficult to reproduce the data.

Reply: Similarly to the previous item, in order to address this comment we have added the details from the appendices in order to make the description clear. We have also added the main results of the manuscript before going to detailed analysis.

Weaknesses 3 - Are all the data shown necessary to reach the conclusions of the paper?

Report: Also the scientific content which seemingly is sufficiently high requires reconsideration, see requested changes and the comments below.

Requested changes e/Authors should limit the numerical results presented in the appendices to a subset needed for the confirmation of the work claims and its conclusions. f/For example what is the relevance of the "spectrum of hopping" (appendix D) as well as Ioffe-Regel criterion (F) for the content of the paper? Either discuss this issues and their relevance to the main body of the paper or remove them for clarity.

Reply: Before submitting to SciPost being in contact with several readers, we faced some problems of the previous versions of the manuscript, related to insufficient data to reach the main conclusions of the manuscript. Therefore, we had to produce additional data and consider more physical variables to confirm our claims. As a result, the version submitted to SciPost had so much data. In the revised version, we make the usage of different physical probes balanced and motivated by the concrete aspects of the problem. Appendices C.1-C.3 have been removed from the text. The corresponding references have been removed as well.

Weaknesses 4 - Interpretation of the numerics seems controversial

Reply: Here we gratefully disagree with the referee 1. The main claim of the reentrant localization mediated by the anisotropy parameter $\beta$ is consistent with our numerics, while the significant finite-size effects (known to be drastic in 2D systems) bring difficulties to the finite-size analysis of the system. In the revised version, we have clarified this point by emphasizing the main crucial points of the analysis and oppose them to the technical issues (such as finite-size effects).

Requested changes a/In the caption of Fig.1 it is written that the transition occurs at r≃0.47 with r defined in Eq.(4) while in Fig.2 or Fig.3 ⟨r⟩ is used to describe the vertical axis; please unify the notation. Strictly the level statistics is given as a set of rn,1 values with the average r. This should be stated around (4).

Reply: In the revised version, we have unified the notations of the average ratio statistics and also clarified the difference between energy-resolved r-statistics and the one averaged over the spectrum.

Requested changes b/it is not frequent to call an earlier paper of one of coauthors as a pioneering paper'' (p.4). Maybe the use of early'' instead of pioneering'' would sound better?

Reply: In the revised version, we have changed “pioneering” by “earlier principle” as the reference [18] established the field of long-range correlated models which is now boosted by a bunch of further papers.

Requested changes c/The level statistics ratio analysis should be performed in energy dependent style as described in the report. d/Similar comments affect other measures used in the paper/appendix.

Report: Let me exemplify the problem with the analysis of level spacing ratio (gap ratio) statistics. For systems with energy dependent mobility edge the safe, standard approach is to find the mean ratio, say r, in narrow energy intervals (see e.g. Luitz et al. 2015, PRB) averaging over different disorder realisations. It seems to be indeed done, somehow, in Fig.2b) where r is plotted as a function of energy. While for β=1,2 r seems not to be dependent on energy and shows signs of localized and extended phases, respectively, the intermediate β=1.5 case seems to show significant energy dependence. While in the caption we can read that data are for L=250, one cannot find out the width of energy window used for this plot (each yellow point for ⟨r⟩ corresponds, I believe, to some tiny energy interval). It is written that vertical dashed lines provide "the position of FSME extracted from finite-size data for r-statistics". But in fact the dashed lines point to the edge of the spectrum where, at low energies, density of states (DOS a) panel) significantly drops indicating the edge of the system. This cannot be the position of the mobility edge! It is not written in the caption which "r-statistics" is being considered. If this is the one shown in Fig.3 then it is performed badly. As written, energies from [−W/2,W/2] interval, i.e. "95% of all the states" are taken for analysis. But this is simply wrong, as mean rr depends on energy, see Fig.2b) - the authors are again referred to e.g. Luitz et al. seminal paper. The proper way of doing the analysis, let me repeat, would be to devide the energy interval [−W/2,W/2] in which the bulk of the spectrum is contained into say 20 or 30 intervals and make the finite size analysis ala Fig.3 in each of the interval separately getting the critical β(E) value depending on energy. Authors are aware of the strong energy dependence of eigenstates, in particular in Fig.8 they study f(α) around energy E=5.

Reply: It seems that this issue is both the confusion and the drawback of poor presentation. In the previous version of the manuscript, indeed, Fig. 2(b) showed the energy-resolved r-statistics and indeed to the right of the vertical dashed line it barely depends on the energy for $\beta=1$ and $2$. The critical point $\beta=1.5$ for this plot shows some inhomogeneity across the spectrum which can be the issue of the real mobility edge. As for the finite-size mobility edge, we emphasize once again its finite-size nature and the applicability only to the localized phase ($\beta=1$ in this case). We call it the mobility edge as it separates the ergodic high-negative-energy states from the bulk of the spectrum where all the states are localized. Figs. 9-10 show that this is just a finite-size effect as the fraction of these ergodic high-negative energy states goes down according to the analytical prediction (which is just as a power of the logarithm of the system size). As for the position of the dashed vertical line “at the edge of the spectrum”, unlike the usual short-range single-particle or many-body disordered systems, where at the edge of the spectrum there are some localized states, in the dipolar (long-range) systems there are high-energy ergodic states. It is these states, which provide the localization of the measure one of the rest bulk states beyond the convergence of the locator expansion. To sum up, - we gratefully disagree with the referee about the finite-size mobility edge at “the edge of the spectrum” due to its different nature drastically affecting the considered effect of anisotropy-mediated localization; - we claim that we considered energy-resolved r-statistics properly (the sliding window average over energy in Fig. 2 is given by 50 adjacent states) and addressed the main criticism of the referee 1 already in the previous version of the manuscript; - we indeed focused on the bulk of the spectrum (95% in our system sizes within the energy interval [-W/2, W/2]) which is not affected by the presence of the finite-size mobility edge and show homogeneous behavior away from the critical point $a=\beta$. - Therefore we do not see a reason to redo the analysis in terms of separating into 20-30 energy intervals as the disorder-averaged energy-resolved data in all these cases shows homogeneity in the spectral bulk [-W/2, W/2]. - Except Fig. 2, yellow lines in Fig. 6, and red lines in Fig. 7, we do not consider in details the critical states at $a=\beta$, but focus mostly on the localized and ergodic states at $a<\beta$ and $a>\beta$, respectively. - The critical behavior at $a=\beta$ needs us to consider smaller energy intervals due to the energy dependence of physical quantities (shown by yellow lines in Fig. 2). In the revised version, we have unified the notations of the average ratio statistics and also clarified the difference between energy-resolved r-statistics and the one averaged over the spectrum. We have also clarified the main issue with the energy-resolved physical quantities and our focus on the bulk of the spectrum in the rest of the work.

Report: By the way, a similar analysis could be done for different disorder amplitude W values as the manuscript considers a single choice W=20.

Reply: The disorder dependence in long-range disordered systems is known to be subleading and related to the finite-size effect (see, e.g., review [30]). Only the critical multifractal states can be affected by this. The origin of this insensitivity is based on the standard limiting cases. On one hand, $a=\infty$ corresponds to a short-range Anderson model in low dimension $d\leq 2$ for which {\it any} finite disorder localize all the states. On the other hand, $a=0$ corresponds to the Wigner-Dyson random matrices which are known to show ergodic behavior for any finite diagonal disorder (or even scaling with the matrix size slower than $N^{1/2}$, see [38]). To sum up, even for our 2d dipolar model the increase of the diagonal disorder may change only finite-size effects, especially in the localized phase: decreasing them for $a>2$ and increasing them for $a<2$ (see the RG analysis where $\langle l^2 \rangle \sim W^2 R^{2(a-d)}$ grows with $W$. Therefore, we considered the effect of the disorder $W$ only for the critical point $a=beta=2$ in Fig. 7, where it is relevant (see also [36] for more details). In the revised version, we have discussed the subleading effects of the disorder amplitude $W$.

Report: Panel 2c) shows the dimension D2 which is both strongly energy dependent for β=1.5 (again - how yellow curve is obtained?). Its noninteger value suggests a fractal character. Multifractality requires, however, that Dq change with q. The statement in the caption that panel b) and c) "show localized, multifractal and ergodic eigenstate properties in the spectral bulk" is thus wrong. In particular panel b) shows mean r which is the eigenvalues property and is not linked with multifractality.

Reply: In Fig. 2 we show that in the spectral bulk the states are ergodic (for $\beta=2>a=1.5$), critical ($\beta = a = 1.5$), and localized ($\beta=1<a=1.5$) properties. As usual in the disordered models, the spectral statistics follows the localization and ergodicity breaking. Of course, there are exceptions of highly-correlated models when the level statistics can show different behavior with respect to the eigenstate properties (see, e.g., [16, 26-27] where localization in the momentum space corresponding to the ergodic eigenstates in the coordinate basis gives Poisson statistics or even [38, 63-65] where r-statistics is of Wigner-Dyson form for fractal eigenstates). However, the common wisdom is that the r-statistics is a proper measure of ergodicity-breaking and localization (see, e.g., the above mentioned by the Referee Luitz et al. 2015, PRB or [33]), which can be used in combination with other measures. In this work we are not interested in distinguishing multifractality from fractality at the critical point (however, yellow lines in Fig. 4 and red lines in Fig. 5 provide the required information confirming multifractal nature of the states). In the revised version of the manuscript, we clarify our claim of ergodic (for $\beta=2>a=1.5$), critical ($\beta = a = 1.5$), and localized ($\beta=1<a=1.5$) properties in Fig. 2 and discuss the role of the amplitude of the diagonal disorder $W$ in long-range systems.

Report: Appendices contain a number of results that are not used or referred to in the main text. I believe a selection should be made validating the claims of the paper (or showing its limitations). They are described sometimes not precisely. While, as mentioned above f(α) is analysed around E=5 what about the following Dq analysis?

Reply: The spectrum of fractal dimensions $f(\alpha)$ provides all the required information about the fractal dimensions, as $(q-1) D_q$ is just the Legendre transform of $f(\alpha)$, see Eq. (11). Therefore, following the previous recommendation of the referee 1 to “limit the numerical results presented in the appendices to a subset needed for the confirmation of the work claims and its conclusions” we prefer not to add more detailed information on the points being away from the main claims of the paper.

Report: In Fig.9 a comparison with power-law random banded matrices is shown but is the power law assumed similar to that of the dipole model (1)? In the caption we read about 2d power-law random banded model - how 2d model is constructed? The referee (and [26] cited) knows only a standard PLRBM model where no dimensional structure is present. The precise definition of PLRBM used should be given in 2 lines explaining e.g. what is the sense of W in PLRBM?

Reply: We agree with the Referee 1 that there is no consideration of d-dimensional PLRBM model in [29, 30]. However, taking into account the generic RG consideration from [22-24], one can simply generalize the results for d-dimensional PLRBM with the Anderson transition at $a=d$ by simple change of the 1d distance |m-n| between two sites m and n by d-dimensional one. In the revised version of the manuscript, we have clarified the definition of d-dimensional PLRBM used for the comparison in Fig. 4. The role of the diagonal disorder $W$ in such disordered long-range models is a finite-size effect (except for the critical point $a=d$) and discussed in the reply to “Requested changes d”.

Report: Is energy dependence included in the analysis of IPR in Fig.10? Is energy dependence included in the higher order gap ratio statistics that appears in Sec.C of the Appendix and fig.15? Let me stop here asking the questions about the wealth of the material scattered throughout appendices..

Reply: In the caption of Fig. 10 it is written explicitly that we show “Inverse participation ratio sorted in increasing order versus renormalized state index”. The eigenvalues are not involved, but IPR is energy-resolved. If not written explicitly, we consider the numerical data in the bulk of the spectrum which is homogeneous in the energy interval [-W/2,W/2]. In the revised version of the manuscript, we have clarified it from the very beginning.

---

## Round 2 · Referee Report · Anonymous (Referee 2) · 2021-11-27

Strengths

  1. Addresses an interesting topic using appropriate methods.

Weaknesses

  1. The setup is a little artificial. The angle-dependence makes sense for the regular dipolar interaction, but for the generalized dipolar interaction (realized using ion traps or cavities) there is no reason to expect this angle-dependence.

Report

I think this is a worthwhile addition to the literature. It builds on some previous results of these authors, showing that for very long-range interactions localization of typical states can sometimes be protected essentially by the way low-momentum states 'detach themselves' from the bulk of the spectrum. Long-distance interactions between wavepackets are mediated by these low-momentum states and are therefore suppressed relative to a naive locator expansion. Whether this phenomenon happens or not depends on the sign structure of the long-range interaction, and the dipolar "angle" is a way of tuning this sign structure.

I find this work conceptually quite interesting although from an experimental point of view the setup is highly artificial. But perhaps the authors can comment a little more on what they expect to happen with real dipoles in 3D.

Requested changes

  1. Be more explicit about what is meant by the generalized dipolar interaction.
  2. Add a little more detail from the appendices into the main text on the RG section.

  • validity: top
  • significance: high
  • originality: high
  • clarity: good
  • formatting: reasonable
  • grammar: excellent

Author:  Ivan Khaymovich  on 2022-06-28  [id 2614]

(in reply to Report 2 on 2021-11-27)
Category:
answer to question

We are very grateful to the Referee 2 for high evaluation of our manuscript, topic to which it is devoted and the method used. Below we present the point-to-point reply to the comments of the Referee.

**Report
I think this is a worthwhile addition to the literature. It builds on some previous results of these authors, showing that for very long-range interactions localization of typical states can sometimes be protected essentially by the way low-momentum states 'detach themselves' from the bulk of the spectrum. Long-distance interactions between wavepackets are mediated by these low-momentum states and are therefore suppressed relative to a naive locator expansion.**
Reply:
We thank the referee 2 for a careful reading of the manuscript and for nicely summarizing the results.

**Whether this phenomenon happens or not depends on the sign structure of the long-range interaction, and the dipolar "angle" is a way of tuning this sign structure.**

Reply:
Here we would like to clarify the point: the sign-structure of the long-range potential (if someone thinks about something like staggered potential) plays no role here as it just changes the phase of the eigenstates by alternating sign with the distance. What indeed matters is the sign-structure of the spectrum of the hopping.

**I find this work conceptually quite interesting although from an experimental point of view the setup is highly artificial. But perhaps the authors can comment a little more on what they expect to happen with real dipoles in 3D.**

**Weaknesses 1. The setup is a little artificial. The angle-dependence makes sense for the regular dipolar interaction, but for the generalized dipolar interaction (realized using ion traps or cavities) there is no reason to expect this angle-dependence.**

**Requested changes 1. Be more explicit about what is meant by the generalized dipolar interaction.**

Reply:
We thank the Referee for pointing out this issue. We agree that the realization of the dipole-dipole interaction with a generalized (smaller) power-law exponent in a regular dipolar system would be very helpful, however to the best of our knowledge only the powers $a=3$ and $a=6$ can be realized in the standard dipolar systems (like Rydberg atoms). In this sense, the direct application of the electric field in such systems is limited to help experimentalists to observe anisotropy-mediated localization.
Nevertheless, in ion trap and ultra-cold atomic setups as well as in 2d photonic crystals near a Dirac point and 2d polaritons, where smaller powers $0<a<2$ can be realized, one can find the analogous effects of anisotropy similar in spirit to the dipole orientation.
In the revised version of the manuscript, we clarify the experimental feasibility of our results in the introduction.

**Requested changes 2. Add a little more detail from the appendices into the main text on the RG section.**
Reply:
In the revised version of the manuscript, we have restructured the main text structure by adding more information from the appendices, including the one mentioned by the referee 2.

---

## Round 3 · Referee Report · Anonymous (Referee 1) · 2022-8-18

Strengths

1 - 2D Anderson transition for the effective long-range tunneling model coming from diluted tilted dipoles is studied apparently showing an interesting reentrant transition
2 - Extensive numerical data support the analysis

Weaknesses

Some numerically based conclusions are not convincing

Report

I believe the manuscript has been significantly improved with respect to the previous version. It contains significant new results on the rich 2D model, some of them not obvious at first glance. The paper combines pure numerical studies with renormalization group analysis. I hope that the authors may take into account the suggested changes as listed below.

Requested changes

  1. Considering the gap ratio statistics the authors rejected my suggestion to make the more detailed energy dependent study arguing that the energy dependence shows only close to $\beta \approx a$ transition. While this is not documented in the figures, it is clearly stated in p.8 so I withdraw my suggestion. Still considering gap ratio statistics Fig.3 shows that finite size scaling of two crossings present in case (c) and (d) yields different $\beta_{AT}$ values. On the other hand, as mentioned in p.5 there is a symmetry relating $0<\beta<2$ interval with $\beta>2$. Is it not this symmetry which leads to the second crossing in panels (c) and (d)? Can one be more quantitative about the relation between the corresponding $\beta_{AT}$ values (1.45 versus 3.1 for panels g,h) ? May be we learn more about the consistency of the finite size scaling from such a comparison?
  2. This referee is entirely lost with Fig.4. The caption and the horizontal axis says $D_2$ versus $a$. On the other hand the points both yellow and blue are denoted by $a=0$. Is it not a bit inconsistent? Caption says yellow squares are for $W=10$ while in the figure $W=20$ is indicated. Something is simply wrong here.
  3. Fig.5 seems to show that three color lines for different values of $L$ yield the extrapolated black curve which is far far above finite size results. Is such an extrapolation not too courageous? What are the errors of this procedure?
  4. The authors are asked to review and correct some formulae. In (17) the equality with Dirac delta function is of course wrong for $t>0$, also imaginary unit "i" is missing from exponents in (17) and (18).
  5. Few places require editing. Eg. p. 3 "By tailor the optical forces" should be "Tailoring the optical forces"; Double dot ending para after (2) in p. 5 is not needed etc...

  • validity: good
  • significance: good
  • originality: good
  • clarity: ok
  • formatting: reasonable
  • grammar: good

Author:  Ivan Khaymovich  on 2022-08-28  [id 2764]

(in reply to Report 1 on 2022-08-18)
Category:
answer to question

We are very grateful to the Referee 1 for his/her detailed second review of our manuscript. Below we present the list of amendments resulting from the comments of the Referee. The revised version of the manuscript, with the changes marked by red font and yellow highlights, is attached to the reply.

Weaknesses - Some numerically based conclusions are not convincing.

Reply: We believe that in the revised version of the manuscript the numerically-based conclusions, consistent with the analytical predictions, are convincing for the Referee 1.

Report: I believe the manuscript has been significantly improved with respect to the previous version. It contains significant new results on the rich 2D model, some of them not obvious at first glance. The paper combines pure numerical studies with renormalization group analysis. I hope that the authors may take into account the suggested changes as listed below.

Reply: We thank the Referee 1 for high evaluation of our revised version and provide the reply below.

Requested changes 1. Considering the gap ratio statistics the authors rejected my suggestion to make the more detailed energy dependent study arguing that the energy dependence shows only close to $\beta = a$ transition. While this is not documented in the figures, it is clearly stated in p.8 so I withdraw my suggestion.

Reply: We thank the referee for this.

Requested changes : 1. (continuation) Still considering gap ratio statistics Fig.3 shows that finite size scaling of two crossings present in case (c) and (d) yields different $\beta_{AT}$ values. On the other hand, as mentioned in p.5 there is a symmetry relating $0<\beta<2$ interval with $\beta>2$. Is it not this symmetry which leads to the second crossing in panels (c) and (d)? Can one be more quantitative about the relation between the corresponding $\beta_{AT}$ values (1.45 versus 3.1 for panels g,h) ? May be we learn more about the consistency of the finite size scaling from such a comparison?

Reply: Yes, indeed, the pairs of crossings in panels (c) and (d) of Fig. 3 are related to each other by the exact symmetry (3) of the model: the fact that the critical exponents are the same in each of the pairs implicitly confirms this. In the revised version, in order to make the symmetry clear, we have added the clarifying phrase into the text: “Note that the pairs of crossings $\beta_{AT}$ in Fig. 3(c, d) are related to each other with respect to the symmetry (3) within the above mentioned error bar, while the critical exponents are just the same.”. In addition, we have added the error bars for $\beta_{AT}$ given by $\pm 0.05$ for $\beta<2$ and by $\pm 0.1$ for $\beta>2$.

Requested changes : 2. This referee is entirely lost with Fig.4. The caption and the horizontal axis says $D_2$ versus $a$. On the other hand the points both yellow and blue are denoted by $a=0$. Is it not a bit inconsistent? Caption says yellow squares are for $W=10$ while in the figure $W=20$ is indicated. Something is simply wrong here.

Reply: We thank the Referee 1 for pointing out these typos. In the revised version of the manuscript we have modified both the figure and the caption accordingly.

Requested changes: 3. Fig.5 seems to show that three color lines for different values of L yield the extrapolated black curve which is far far above finite size results. Is such an extrapolation not too courageous? What are the errors of this procedure?

Reply: Yes, indeed, Fig. 5 shows the extrapolation of the finite size spectrum of fractal dimensions $f(\alpha, L)$ to the infinite system size. As usual (see, e.g., [16, 17, 37]) with $f(\alpha)$ the main finite size correction of $f(\alpha, L)$ is given by the vertical shift (i.e. the weakly $L$-dependent prefactor in the probability distribution of $\alpha$, given by subleading corrections in (12)). As the shape of $f(\alpha, L)$ is practically the same for all the system sizes that we considered, we just get rid of the (unknown) prefactor by extrapolation (12). The deviations of the extrapolated value of the maximum of $f(\alpha)$ from $1$ gives an estimate for the error bar of this procedure.

Requested changes: 4. The authors are asked to review and correct some formulae. In (17) the equality with Dirac delta function is of course wrong for $t>0$, also imaginary unit "i" is missing from exponents in (17) and (18). 5. Few places require editing. Eg. p. 3 "By tailor the optical forces" should be "Tailoring the optical forces"; Double dot ending para after (2) in p. 5 is not needed etc…

Reply: We thank the Referee 1 for pointing out the misprints and correct them accordingly in the revised version.

Attachment:

Text_for_reply1_SciPost.pdf

---

## Round 3 · Author Response

Dear Editor,

Hereby we resubmit our manuscript for consideration in SciPost Physics.
Following the recommendations of both referees, we have restructured the manuscript and implemented corrections.
Please see the list of changes below and the point-to-point replies to the referees in the previous version as comments.

Looking forward to your decision.

Sincerely yours,
Xiaolong Deng, Alexander L. Burin, and Ivan M. Khaymovich

---

## Round 3 · List of Changes

1. We clarify the experimental feasibility of our results and added some additional examples from the former Appendix G.
2. We have restructured the main text by adding more information from the appendices:
a. Section 2 has been renamed to “Model and its symmetry”.
b. Section 3 has been renamed to “Overview of the numerical and analytical results”: now it contains Fig. 2 and the reference to all the results from Sec. 4 and 5.
c. Section 4 “Numerical results” contains the following subsections:
i. Sec. 4.1. “Finite-size flow of the ratio $r$-statistics”, consisting of Fig. 3 and the first part of the former Appendix A.1.
ii. Sec. 4.2 “Eigenstate properties: multifractal analysis and wave-function spatial decay”, consisting of the results for the spectrum of fractal dimensions $f(\alpha)$, fractal dimensions $D_q$, and the wave-function spatial decay: Figs. 4-7 and the former Appendix A.2.
iii. Sec. 4.3 “Wave-packet dynamics. Return probability” consists of Fig. 8 and the former Appendix B.2.
iv. Sec. 4.4 “Finite-size mobility edge and the fraction of ergodic states” consists of Figs. 9-10 and the second part of the former Appendix A.1 and Appendix A.3.
d. Section 5 “Analytical methods and results”, first, explains with the help of the new Fig. 11 the main idea of the localization in the spectral bulk due to the presence of the high-energy ergodic states. It contains the following subsections
i. Sec. 5.1 “Main idea of the renormalization group analysis”, including the former Appendix E, and
ii. Sec. 5.2 “Ioffe-Regel criterion for the fraction of high-energy ergodic states”, including the former Appendix F.
e. Appendix A.1 has been moved to Secs. 4.1, 4.4.
f. Appendix A.2 has been moved to Sec. 4.2.
g. Appendix A.3 has been moved to Sec. 4.4.
h. Appendix B.1 has been renamed to Appendix B.
i. Appendix B.2 has been moved o Sec. 4.3.
j. Appendix C (before C.1) has been renamed as the new Appendix A.
k. Appendices C.1-C.3 have been removed.
l. Appendix D has been renamed as the Appendix C.
m. Appendices E and F have been moved to the first and second subsections of Sec. 5.
n. Appendix G has been partly merged to the introduction.
3. We have separated the description of finite-size effects and the main crucial points of our numerical analysis.
4. We have unified the notations throughout the text.
5. We have clarified the sliding window average, used to produce the results in Fig. 2, and also the difference between energy-resolved r-statistics and the one averaged over the spectrum.
6. We have also clarified the main issue with the energy-resolved physical quantities and our focus on the bulk of the spectrum in the rest of the work, where the properties are homogeneous away from the critical points $\beta = a$.
7. We have discussed the subleading effects of the disorder amplitude $W$ with respect to the exponent $a$ of the power law and the anisotropy parameter $\beta$.
8. We have clarified our claim of ergodic (for $\beta=2>a=1.5$), critical ($\beta = a = 1.5$), and localized ($\beta=1<a=1.5$) properties in Fig. 2.
9. We have clarified the definition of d-dimensional PLRBM used for the comparison in Fig. 4.
10. We have improved the caption of Fig. 10 of the energy-resolved IPR sorted by increasing amplitude.
11. Fig. 1(a) has been replaced.
12. Fig. 3 has been combined from the former Figs. 3 and 6.
13. Fig. 9 has been combined from the former Fig. 7 and the inset to Fig. 3(b).
14. Fig. 11 has been added for the clarification of the analytical idea.
15. Several relevant references have been added [27, 47, 48] or updated [36].
16. Due to removing Appendices C.1-C.3 and G, the reference [65-74, 76-77] have also been removed.
17. Several clarifications, minor amendments, and corrections of typos have been done throughout the manuscript.

---

## Editorial Decision

resubmitted